# STREAM: Spatio-TempoRal Evaluation and Analysis Metric for Video Generative Models

**Pum Jun Kim, Seojun Kim, Jaejun Yoo**
Ulsan National Institute of Science and Technology
{pumjun.kim,seojun.kim,jaejun.yoo}@unist.ac.kr

## Abstract

Image generative models have made significant progress in generating realistic and diverse images, supported by comprehensive guidance from various evaluation metrics. However, current video generative models struggle to generate even short video clips, with limited tools that provide insights for improvements. Current video evaluation metrics are simple adaptations of image metrics by switching the embeddings with video embedding networks, which may underestimate the unique characteristics of video. Our analysis reveals that the widely used Fréchet Video Distance (FVD) has a stronger emphasis on the spatial aspect than the temporal naturalness of video and is inherently constrained by the input size of the embedding networks used, limiting it to 16 frames. Additionally, it demonstrates considerable instability and diverges from human evaluations. To address the limitations, we propose STREAM, a new video evaluation metric uniquely designed to independently evaluate spatial and temporal aspects. This feature allows comprehensive analysis and evaluation of video generative models from various perspectives, unconstrained by video length. We provide analytical and experimental evidence demonstrating that STREAM provides an effective evaluation tool for both visual and temporal quality of videos, offering insights into area of improvement for video generative models. To the best of our knowledge, STREAM is the first evaluation metric that can separately assess the temporal and spatial aspects of videos. Our code is available at STREAM.

## 1 Introduction

"Measure what is measurable, and make measurable what is not so." This quote by Galileo Galilei reflects the underlying philosophy of many scientific advancements. In a similar vein, Peter Drucker famously stated, "If you cannot measure it, you cannot manage it." Recent breakthroughs in powerful generative models (Karras et al., 2020; Sauer et al., 2022; Rombach et al., 2022) have ushered in an era of generating highly realistic images, largely due to insightful evaluation metrics (Heusel et al., 2017; Salimans et al., 2016; Kynkäänniemi et al., 2019; Naeem et al., 2020; Kim et al., 2023) that have shaped their progress. However, as we pivot to video generative models, there is a pronounced gap. Many of contemporary video generative models often struggle to generate even concise video clips. Numerous recent studies have proposed a variety of solutions, ranging from crafting new architectures or modules to oversee entire video frames (Tulyakov et al., 2018; Li et al., 2019), to the introduction of regularization techniques aimed at ensuring temporal consistency (Sun et al., 2020). While these innovations show promise, they come with an inherent limitation: the necessity for a measurable and reliable metric to gauge the actual extent of their intended improvements. The absence of comprehensive metrics for analyzing and evaluating these models leaves researchers without a clear guide to model enhancements, hindering further progress.

At first glance, using image evaluation metrics on each frame of a video and computing an average might seem sufficient. This method, however, neglects an essential aspect: videos are more than mere collections of individual images. For a video to feel genuinely natural, a model needs to not only generate high-quality images for each frame but also to ensure that the content is temporally consistent, fluid, and seamless across frames. Seeking for a more objective assessment, most current video generative models use Video Inception Score (VIS) (Saito et al., 2020) and Fréchet Video Distance (FVD) (Unterthiner et al., 2019) as their go-to metrics. These are essentially derivatives of

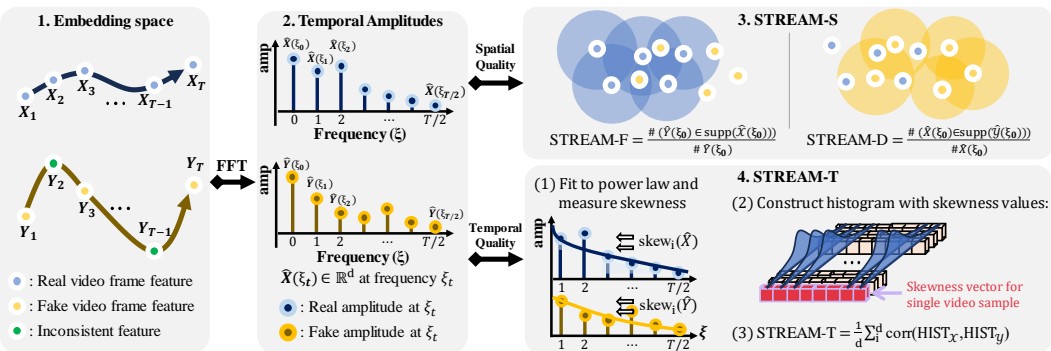

Figure 1: An illustration of the proposed evaluation pipeline. We use image embedding space to evaluate video regardless of its length and to consider the spatial and temporal aspects of video independently (Section 2.2). Then, we use the Fast Fourier Transform (FFT) along the temporal axis of frame features to capture the variation over time for evaluation and to utilize the average at frequency zero for spatial evaluation (Section 2.1). Finally, we calculate STREAM-S and STREAM-T to evaluate the video generative models (Section 2.4 and 2.3).

the earlier Inception Score (IS) (Salimans et al., 2016) and Fréchet Inception Distance (FID) (Heusel et al., 2017), initially designed for image generative models. This adaptation entails replacing the core embedding network with one optimized for videos such as Inflated 3D Convnet (I3D) (Carreira & Zisserman, 2017). However, based on our in-depth analyses, we find that there are considerable gaps in the existing metrics. For instance, the widely used FVD predominantly focuses on the spatial attributes of video content, with a significant underemphasis on consistently evaluating temporal flow. This becomes a pressing concern as video generative models advance. Moreover, as these models develop capabilities to generate longer videos, FVD remains constrained to evaluating videos with a mere 16 frames, heavily dependent on the constraints of the utilized embedding network; *e.g.*, the input size that the network can handle. Therefore, to develop properly functioning video generative models, it is essential to evaluate videos of varying lengths, from short video clips to longer videos, and to have metrics that can assess both spatio-temporal quality separately.

Addressing the identified limitations, we propose Spatio-TemproRal Evaluation and Analysis Metric (STREAM), a new metric designed to separately assess the temporal naturalness (STREAM-T) and the realism and diversity of videos (STREAM-S). STREAM-T evaluates the congruity in the overall trend of temporal changes between consecutive generated video frames in comparison to those observed in real ones. STREAM-S comprises two evaluation components: STREAM-F, dedicated to analyzing the fidelity of videos, and STREAM-D, focused on evaluating the diversity within videos. Through a series of carefully designed experiments, we demonstrate the efficacy of STREAM in analyzing and evaluating the spatio-temporal aspects of video generative models. Our findings reveal the prevailing challenges in current video generative models, showcasing their struggle to generate video frames that are both realistic and diverse, particularly when the video length increases.

**Our contributions are summarized as follows:**

- **Uniqueness:** We propose STREAM, a new video evaluation metric. STREAM is the first evaluation metric that can separately assess the temporal and spatial aspects of videos.

- **Boundness:** STREAM stands out as a unique metric offering granular evaluation of videos and providing bounded values, a notable advancement over FVD which only allows comparative assessment.

- **Universal Applicability:** STREAM applies to all video generative models. Furthermore, STREAM assesses video regardless of video length, offering a versatile solution to analyze a broad spectrum of videos.

- **Explainability:** We demonstrate the efficacy of STREAM under various scenarios, revealing the challenges and unsolved problems faced by current models, particularly in long video generation.

## 2 SPATIO-TEMPORAL EVALUATION AND ANALYSIS METRIC (STREAM)

STREAM separately evaluates the spatial and temporal aspects of generated video dataset (Figure 1). We first describe the embedding network for evaluation in Section 2.2 and introduce STREAM-T, a metric for assessing the temporal aspect of videos in Section 2.3. Next, we introduce STREAM-S, a metric for assessing the spatial aspect of videos in Section 2.4.

### 2.1 NOTATION

Let real and fake video datasets be denoted as $\mathcal{X}$ and $\mathcal{Y}$, respectively. We refer to the support of $\mathcal{X}$ as supp($\mathcal{X}$). We denote the a real video as $X = \{X_1, X_2, ... X_T\} \in \mathcal{X}$ where $T$ is a number of frames or time. The projected real video $X_{proj}$ through image embedding network $f(\cdot)$ is written as $X_{proj} = f(X) = \{f(X_1), f(X_2), ..., f(X_T)\} \in \mathbb{R}^{d \times T}$, where $d$ is the feature dimension. Throughout this paper, we refer to $X$ as $X_{proj}$ for simplicity. After applying FFT to $X$ along time axis, the Fourier amplitude is defined as $\hat{X} = \{\hat{X}(\zeta_0), \hat{X}(\zeta_1), ..., \hat{X}(\zeta_{T/2})\} \in \mathbb{R}^{d \times T/2}$, where $\zeta$ is a frequency with the order of $\zeta_0 < \zeta_1 < ... < \zeta_{T/2}$ and $\hat{X}(\zeta_0 = 0)$ is a mean amplitude by definition. We use the notations for $\mathcal{Y}$ in the same manner as $\mathcal{X}$.

### 2.2 EMBEDDING NETWORK

Evaluation metrics for generative models heavily rely on the embedding networks trained on large-scale dataset. This is because, embedding networks are known to extract features that align with human perceptual quality (Theis et al., 2015) and facilitate the handling of high-dimensional data. The choice of an embedding network is thus critical when assessing the quality of generated videos. However, existing video embedding networks are often unsuitable for effective evaluation yet, experiencing the intrinsic challenges of video embedding. These challenges include either the imperfect encoding of temporal information in video data (Li et al., 2022; Wang et al., 2022) or the mixed encoding of spatio-temporal information coupled with rigid constraints on the number of input video frames (Wang et al., 2023), significantly impeding long-term applicability and efficacy. To circumvent these limitations, we utilize a proficient image embedding network proposed by Caron et al. (2020) as the foundation for our evaluation metric. By encoding each video frame independently, this approach enables separate consideration of spatial and temporal aspects of videos, unhindered by the number of video frames.

### 2.3 **STREAM-T:** EVALUATING THE TEMPORAL FLOW OF VIDEOS

To capture the temporal trend of independent frame features, we estimate a power law distribution of frequency amplitudes and assess its skewness. By aggregating the skewness of both real and fake features and measuring their correlation, we compute the temporal flow score, "STREAM-T".

**Distribution for video time-series signals** The temporal variations of each feature within the embedding space can be perceived as variations in the amplitude of frequencies, or the spectrum. From this standpoint, one might attempt to compare temporal flow between real and generated data by simply examining the mean or variance of these amplitudes. However, applying these measures directly may introduce bias. This is often due to the dominance of power spectra at low frequencies, a common characteristic observed in natural signals (West & Shlesinger, 1990). Such dominance can obscure the genuine variations present in high-frequency components, leading to inaccurate evaluation of temporal flow differences. This phenomenon is referred to as "$1/f$ fluctuation" of the spectral density, $S(f)$, of a stochastic process, having the form of $S(f) = \text{constant}/f^{\alpha}$ with $f$ representing frequency. To better observe the differences in temporal flow, we transform this power law into the form of a probability distribution. For a given arbitrary feature dimension $k$ and frequency $\zeta_i$ (where $i \in [0, T/2]$), $\hat{X}_k(\zeta_i)$ can be approximated with power law $C \cdot \zeta_i^{-\alpha}$, where $\alpha$ is a coefficient, and $C$ is a normalization constant. Here, the coefficient $C$ is a simple scaling factor, and $\alpha$ plays a role in adjusting the overall slope of the power law. To approximate the parameters $\alpha \in \mathbb{R}$ and $C \in \mathbb{R}$, we solve $\hat{X}_k \approx \tilde{X}_k = C \cdot \zeta^{-\alpha}$ through least square estimation. Then the power law distribution (or amplitude distribution) is defined as $p_{\hat{X}_k}(\zeta) = \frac{C}{K} \cdot \zeta^{-\alpha}$, where $K$ is the normalization constant.

**Skewness of data distribution**   We calculate the difference in temporal flow of real and generated data by comparing the skewness of the power law distribution, $p_{\hat{X}_k}(\zeta) = \frac{C}{K} \cdot \zeta^{-\alpha}$, rather than comparing the mean or variance of Fourier amplitudes. First, for a random variable $X$ with a known probability distribution, the skewness is defined as $\gamma = E[(X - \mu)^3]/\Sigma^3$, where $E[(X - \mu)^3]$ is the third moment of the distribution, and $\Sigma$ is the variance. To estimate the parameters in the above equation, we utilize the widely known method of Moment Generating Function (MGF) (see Appendix A.1 for details). By calculating the second and third moments of $p_{\hat{X}_k}(\zeta)$ using the MGF, we can approximate all the necessary parameter and the skewness of power law distribution is computed as follows:

$$skewness \; \gamma_{X,k} = \frac{E[(\zeta - E[\zeta])^3]}{Var(\zeta)^3} = \frac{\sqrt{K}\sum_\zeta \zeta^{(3-\alpha)}}{\sqrt{C\sum_\zeta \zeta^{(2-\alpha)}}}$$

**Evaluating temporal flow of videos**   As in Algorithm 1, for a given real feature $X$ and fake feature $Y$, we first perform FFT to obtain the real and fake amplitude features $\hat{X} \in \mathbb{R}^{d \times T/2}$ and $\hat{Y} \in \mathbb{R}^{d \times T/2}$, respectively. Through this, we interpret real and fake temporal flows in the form of signals. For each dimension of the amplitude features, $\hat{X}_k \in \mathbb{R}^{1 \times T/2}$ and $\hat{Y}_k \in \mathbb{R}^{1 \times T/2}$ ($k \in [1, d]$), we approximate the power law distribution. This gives us real and fake power law distributions at feature dimension $k$, $f(\hat{X}_k, \alpha_{X_k}, C_{X_k})$ and $f(\hat{Y}_k, \alpha_{Y_k}, C_{Y_k})$, respectively. In this step, we do not utilize the real and fake amplitude values when the frequency is 0, because it leads STREAM-T to respond to both visual and temporal qualities (see Appendix A.3). From the given power law distributions, we measure the real skewness $\gamma_{X,k}$ and fake skewness $\gamma_{Y,k}$ at feature dimension $k$. We repeat this process for all feature dimensions, which results in $\gamma_X$ and $\gamma_Y$.

To compare skewness between the real and fake at all feature dimensions, we construct histograms of real $hist(\gamma_X)$ and fake $hist(\gamma_Y)$ for each feature dimension $d$. Then we calculate the correlation $\rho_{X,Y}$ between the real and fake histograms at all feature dimension $d$. The STREAM-T is defined as the average of the cor-

relation. Note that, We have configured the histogram's bin size to 50. The bin size ranging from 50 to 100 have minimal impact on the performance, but larger bins can induce unstable performance with a limited data size (See Appendix A.11).

---

**Algorithm 1** STREAM-T

**Input**: real feature $X$, fake feature $Y$
**Given**: power law distribution $f(\cdot)$
**for** $X \in \mathcal{X}$ and $Y \in \mathcal{Y}$ **do**
   $\hat{X} \leftarrow FFT(X) \in \mathbb{R}^{d \times T/2}$
   $\hat{Y} \leftarrow FFT(Y) \in \mathbb{R}^{d \times T/2}$
   **for** $k = 1$ to $d$ **do**
      $p_{\hat{X}_k}(\zeta) \leftarrow f(\hat{X}_k, \alpha_{X_k}, C_{X_k})$
      $p_{\hat{Y}_k}(\zeta) \leftarrow f(\hat{Y}_k, \alpha_{Y_k}, C_{Y_k})$
      $\gamma_{X,k} \leftarrow skewness(p_{\hat{X}_k}(\zeta)) \in \mathbb{R}$
      $\gamma_{Y,k} \leftarrow skewness(p_{\hat{Y}_k}(\zeta)) \in \mathbb{R}$
   **end for**
   $\gamma_X \leftarrow \{\gamma_{X,1}, \gamma_{X,2}, ..., \gamma_{X,d}\} \in \mathbb{R}^{1 \times d}$
   $\gamma_Y \leftarrow \{\gamma_{Y,1}, \gamma_{Y,2}, ..., \gamma_{Y,d}\} \in \mathbb{R}^{1 \times d}$
**end for**
$\gamma_\mathcal{X} \leftarrow \{\gamma_X; for \; all \; X \in \mathcal{X}\} \in \mathbb{R}^{n \times d}$
$\gamma_\mathcal{Y} \leftarrow \{\gamma_Y; for \; all \; Y \in \mathcal{Y}\} \in \mathbb{R}^{n \times d}$
# calc histogram for each feature dimension $d$
$HIST_\mathcal{X} \leftarrow hist(\gamma_\mathcal{X})$ and $HIST_\mathcal{Y} \leftarrow hist(\gamma_\mathcal{Y})$
# calc correlation for each feature dimension $d$
$\rho_{\mathcal{X},\mathcal{Y}} \leftarrow corr(HIST_\mathcal{X}, HIST_\mathcal{Y}) \in \mathbb{R}^d$
**return** $\frac{1}{d}\sum_1^d \rho_{\mathcal{X},\mathcal{Y}}$

---

## 2.4   STREAM-S: EVALUATING THE SPATIAL QUALITY OF VIDEOS

**Evaluating fidelity and diversity of images**   In the image generation task, Kynkäänniemi et al. (2019) have proposed improved precision and recall (P&R) metric to separately evaluate the fidelity and diversity of the generated image quality. Given real dataset $\mathcal{X}$ and fake dataset $\mathcal{Y}$, precision and recall are computed as

$$\text{precision}(\mathcal{X}, \mathcal{Y}) := \frac{1}{M}\sum_Y^M f(Y \in \mathcal{Y}, \mathcal{X}), \; \text{recall}(\mathcal{X}, \mathcal{Y}) := \frac{1}{N}\sum_X^N f(X \in \mathcal{X}, \mathcal{Y})$$

where $f(Y \in \mathcal{Y}, \mathcal{X}) = 1$ if $Y \in supp_k(\mathcal{X})$ and otherwise 0, and vice versa. Here, $supp_k(\mathcal{X})$ is the estimated support defined as the union of spheres, where each sphere has a sample point $X$ as its center and a radius equal to the distance to the k-nearest neighbors (k-NN) of $X$. Here, $k$ is the hyper-parameter for $k$-NN, and it is assigned an arbitrary value for use. Likewise, $supp_k(\mathcal{X})$ is defined as a collection of spheres with centers at $Y$ and radius equal to $k$-NN distances of $Y$.

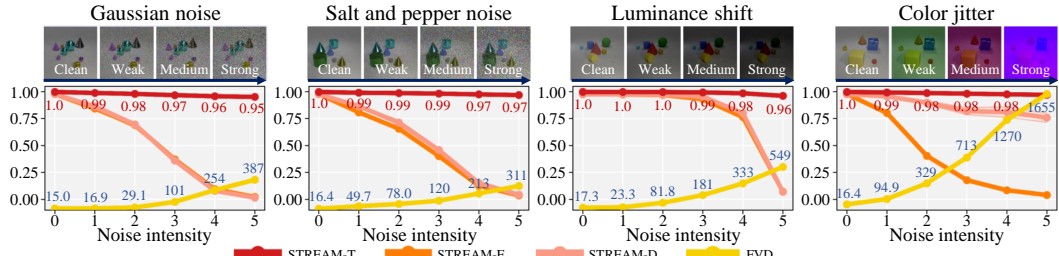

Figure 2: Behavior of STREAM regarding noise affecting the "visual quality". All noise used in the experiment is equally added to entire video frames. Luminance shift decreases the contrast of all frames as the intensity increases. Color jitter is applied by randomly sampling colors for each video sample, thereby affecting the overall color tone of the video.

**Evaluating fidelity and diversity of videos** Simply applying P&R to video dataset has several issues. Firstly, our embedding network provides features for each frame of a video. In the process of P&R support estimation, if we define a sphere with centered at sample point with radius defined by $k$-NN to estimate the support of data, we end up with a support consisting of small spheres that may not even cover the features from a single video sample (See Appendix A.4). Another approach is to use a larger value of $k$ of $k$-NN than the number of video frames, but this approach also does not guarantee that a single sphere includes all the video frame features for a given video. To evaluate the fidelity and diversity of videos, we extend the P&R method for video data using the mean amplitude, which corresponds to the amplitude at frequency of 0. This aligns with the frame-wise average values. Given real and fake mean amplitude features $\hat{X}(\zeta_0) \in \hat{\mathcal{X}}(\zeta_0)$ and $\hat{Y}(\zeta_0) \in \hat{\mathcal{Y}}(\zeta_0)$, respectively, we define the STREAM-F for fidelity and STREAM-D for diversity as

$$\text{STREAM-F} := \frac{1}{M} \sum_{\hat{Y}(\zeta_0)}^{M} f(\hat{Y}(\zeta_0), \hat{\mathcal{X}}(\zeta_0)), \ \text{STREAM-D} := \frac{1}{N} \sum_{\hat{X}(\zeta_0)}^{N} f(\hat{X}(\zeta_0), \hat{\mathcal{Y}}(\zeta_0)).$$

We set the hyperparameter for $k$-NN to $k = 5$.

## 3 EXPERIMENTS

We assess the capability of STREAM to accurately evaluate spatial and temporal aspects of video data. We employ a series of tests involving synthetic toy data and actual samples generated by video generative models to ensure a comprehensive evaluation of the effectiveness and reliability of the proposed metric in various scenarios. In all experiments, we consider a total of 2,048 real and fake data. The results for all metrics are the average of five repeated measurements. In the accompanying figures, the shaded area represents the range of standard deviation.

### 3.1 TOY DATA EXPERIMENT

We use the synthetic CATER dataset (Girdhar & Ramanan, 2020) in our experiments. This ensures a full control over the experimental conditions, allowing us to mitigate the influence of potential confounding factors such as degraded samples, temporally inconsistent samples, and blurry frames, which could compromise the integrity of our experimental setup. The CATER dataset consists of scenarios where multiple objects are positioned against a static background, with only a select few demonstrating movement.

#### 3.1.1 EVALUATING VISUAL QUALITY DEGRADATION IN VIDEO

In this experiment, we consider visual degradation using four types of noise, and the same noise is applied to all video frames. Therefore, the ideal behavior of metrics evaluating the temporal naturalness should not respond to such noise, while metrics assessing the spatial aspects should exhibit a decreasing trend in response to noise intensity. In Figure 2, STREAM-T exhibits robust response to all noise intensities. Meanwhile, STREAM-S consistently decreases and converges to zero as the noise intensity increases. In contrast, FVD shows varying evaluation scores depending on the type of noise, exhibiting a significant difference in these values. Note that there is no upper limit in FVD.

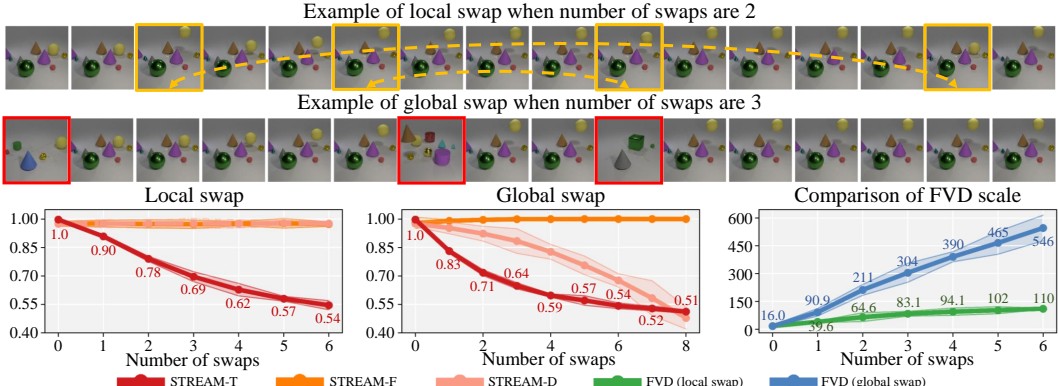

Figure 3: Comparison of the behaviors of STREAM and FVD when changes are introduced to the "temporal flow" of video data. As in the example, local swap involves swapping the orders of two randomly selected frames within the video, while global swap entails exchanging a randomly chosen frame with a frame from another video.

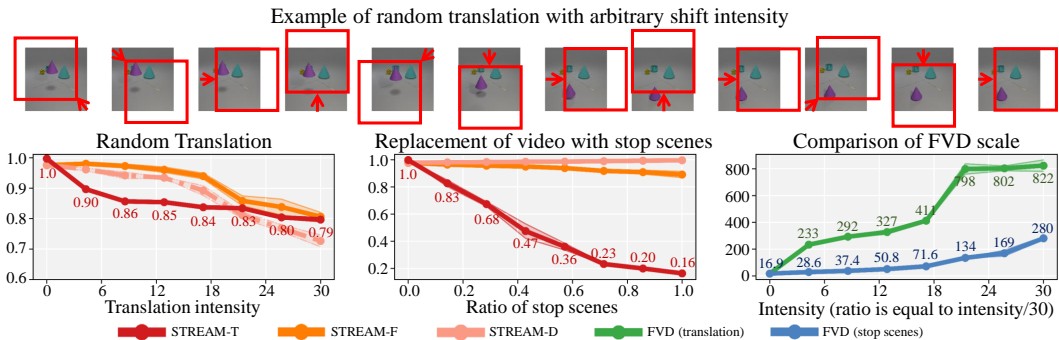

Figure 4: Behaviors of STREAM and FVD in response to various temporal flow modifications. Random translation applies random directional shifts to each video frame. The translation intensity indicates the number of pixels to be shifted in a random direction. Replacement of video with stop scenes replaces a certain proportion of videos in the dataset with video containing only still frames.

### 3.1.2 EVALUATING TEMPORAL FLOW DEGRADATION IN VIDEO

As shown in Figure 3, we manipulate the continuous movement of a video either by swapping the position of two randomly selected frames (local swap) or by integrating frames from another video into the original video (global swap). For example, when local swap occur five times, it leads to distortion where the order of ten video frames is altered. To induce such distortion in global swap experiments, it requires sampling ten frames. An ideal temporal flow metric should accurately detect the degradation in temporal quality in both settings, while an ideal spatial metric should remain unaffected by local swap but should discern the decrease in diversity caused by increased global swap, as this mixes frames between videos, creating more uniform outputs. Our results show that STREAM-T effectively captures the variations in the levels of temporal distortion, and STREAM-S accurately distinguishes the difference between local and global swap. In contrast, FVD shows inconsistent sensitivities across the experiments, being approximately five times more sensitive when there is a reduction in diversity with temporal distortion (global swap).

### 3.1.3 EVALUATING SPATIO-TEMPORAL DEGRADATION IN VIDEO

For "random translation", we randomly determine both the horizontal and vertical directions of movement for each video frame and then shift the frames parallel to these directions by the specified translation intensity. The temporal inconsistencies intensifies as the level of degradation increase. This degradation not only disrupts the temporal flow but also induces spatial distortions because the empty area of the video frame post-shift is filled with the surrounding pixels. The "replacement of

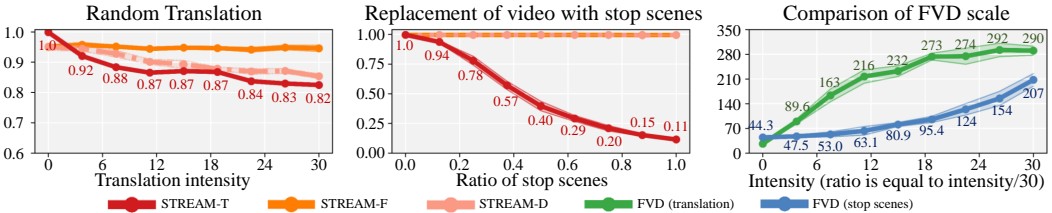

Figure 5: Behavior of STREAM when noise affecting the "visual quality" is applied to the real-world data (UCF-101). All noise used in the experiment is equally added to entire video frames. Color jitter is applied by randomly sampling color filters for each video, which alters the overall color tone of the video.

Figure 6: Temporal distortion experiments using Kinetics-600 data. Identical to Section 3.1.3, this experiment compares the response of each metric when temporal information of video is distorted through (a) random translation, (b) stop scenes. We utilized translation intensity that preserves the realism of the video as much as possible (Figure 10).

video with stop scenes" replaces a certain proportion of videos with static scenes. At a replacement ratio of 1.0, the temporal flow within the video completely disappears. Therefore, an ideal temporal metric should exhibit heightened sensitivity to stop scenes compared to random translations, whereas a spatial metric should exclusively be responsive to random translation. Figure 4 shows that both STREAM-S and STREAM-T respond appropriately in all experiments. In contrast, FVD shows heightened sensitivity to random translation, presenting a response that is more than three times as intense (reaching up to 822) in comparison to situations where temporal information is entirely absent. It is noteworthy that the real dataset does not contain any videos composed entirely of still scenes. This indicates that FVD places a higher emphasis on spatial quality over temporal aspects.

## 3.2 REAL DATA EXPERIMENT

We show the consistent performance of STREAM, using real dataset: Kinetics-600 (Carreira et al., 2018) and UCF-101 (Soomro et al., 2012) datasets, aligning with results from the toy dataset (Section 3.2.1). This consistency underlines the reliability of STREAM in evaluating the real video generative models (Section 3.2.2). Unlike FVD, constrained by its embedding network and unable to assess long video data, STREAM proves effective in evaluating long videos with more than 16 frames (Section 3.2.3).

### 3.2.1 EVALUATING SPATIAL AND TEMPORAL DEGRADATION IN VIDEO

Section 3.1.1 presents an experiment conducted in a synthetic setting where the background remains static, and only simple geometric objects in motion are present. To demonstrate STREAM maintains its effectiveness even when dealing with more intricate data involving moving backgrounds and visual degradation, we have conducted experiments using the UCF-101 dataset (Figure 5). We applied the same gaussian noise, salt and pepper, and color jitter as in the experiment of Section 3.1.1, excluding the simpler degradation of luminance shift. As shown in Figure 5, STREAM-T remains unaffected by all types of visual or spatial noise, which are applied uniformly across all frames. Conversely, STREAM-S, designed to assess spatial aspects, effectively responds to alterations in visual integrity, exhibiting a proportional decrement in response to the escalating intensity of the visual noise applied. In contrast, FVD exhibits significantly different evaluation trends depending on the type of noise introduced. Additionally, we have conducted temporal degradation experiments using

Table 1: Comparison and analysis of video generative models (unconditional). All the models are trained on the UCF-101 dataset generating 16 frame videos with $128 \times 128$ resolution. The numbers in parentheses next to evaluation scores represent the standard deviation of the scores, calculated through five repeated measurements.

|  | VIS ($\uparrow$) | FVD ($\downarrow$) | STREAM-T ($\uparrow$) | STREAM-F ($\uparrow$) | STREAM-D ($\uparrow$) |
|---|---|---|---|---|---|
| MoCoGAN | 16.64 ($\pm$0.09) | 1174.3 ($\pm$36.69) | 0.9683 ($\pm$0.001) | 0.1595 ($\pm$0.023) | 0.0000 ($\pm$0.000) |
| DIGAN | 24.32 ($\pm$0.19) | 763.64 ($\pm$28.82) | 0.9743 ($\pm$0.000) | 0.3101 ($\pm$0.011) | 0.0662 ($\pm$0.005) |
| TATS | 34.39 ($\pm$0.30) | 693.27 ($\pm$21.55) | 0.9832 ($\pm$0.000) | 0.9120 ($\pm$0.011) | 0.0850 ($\pm$0.005) |
| VideoGPT | 30.35 ($\pm$0.55) | 647.75 ($\pm$15.34) | 0.9782 ($\pm$0.000) | 0.7806 ($\pm$0.030) | 0.3272 ($\pm$0.005) |
| MeBT | 64.54 ($\pm$0.51) | 504.21 ($\pm$24.50) | 0.9616 ($\pm$0.001) | 0.7441 ($\pm$0.006) | 0.1852 ($\pm$0.019) |
| PVDM | 60.02 ($\pm$0.82) | 415.70 ($\pm$25.59) | 0.9843 ($\pm$0.002) | 0.6416 ($\pm$0.014) | 0.3112 ($\pm$0.005) |

Table 2: Comparison and analysis of video generative models which produce long video frames (unconditional). All the models are trained on UCF-101 dataset generating 128 frame videos with $128 \times 128$ resolution. sVIS and sFVD denotes the modified version of VIS and FVD measured for every 16 frames using a sliding window. See Appendix A.8 and A.9 for the sample qualities.

|  | sVIS ($\uparrow$) | sFVD ($\downarrow$) | STREAM-T ($\uparrow$) | STREAM-F ($\uparrow$) | STREAM-D ($\uparrow$) |
|---|---|---|---|---|---|
| MoCoGAN | 11.8450 | 1454.1 | 0.3274 | 0.0615 | 0.0000 |
| DIGAN | 18.0075 | 1103.0 | 0.1327 | 0.1206 | 0.2656 |
| TATS | 40.3345 | 1008.0 | 0.0302 | 0.6284 | 0.2104 |
| MeBT | 33.9492 | 948.51 | 0.8265 | 0.6284 | 0.1601 |

Kinetics-600 in the setting analogous to Section 3.1.3. As shown in Figure 6, STREAM behaves as intended, while FVD shows even greater sensitivity to random translation.

### 3.2.2 EVALUATING VIDEO GENERATIVE MODELS

In Table 1, we compare the video generative models using FVD, VIS, and our proposed metrics. The models considered in the experiments are MoCoGAN-HD (Tulyakov et al., 2018), DIGAN (Yu et al., 2022), TATS-base (Ge et al., 2022), VideoGPT (Yan et al., 2021), MeBT (Yoo et al., 2023), and PVDM (Yu et al., 2023). FVD and VIS, offering a singular score, fall short of exposing the varied strengths and limitations of these models. In contrast, our analysis through STREAM reveals that while all considered models maintain reasonable consideration for temporal flow in generating short videos, they typically exhibit low diversity and a deficiency in realism. For example, relying solely on FVD and VIS as benchmarks might give the perception that TATS has mediocre performance compared to the others. However, utilizing STREAM reveals that TATS surpasses other models like MeBT and VideoGPT in terms of realism and temporal naturalness in short video generation, despite its deficiency in diversity, which aligns well with the qualitative analysis (Appendix A.7). To validate the alignment of the results in Table 1 with human perceptual quality, we evaluate the Spearman's rank correlation coefficient between human judgement scores and each metric on realism and temporal coherence. STREAM demonstrates effective representation of human perceptual quality with correlations of 0.9 for realism and 0.6 for temporal naturalness, while FVD remains less effective for both aspects (see Appendix A.6 for further details). Therefore, to truly assess the performance of video generative model, it is important to consider the spatial and temporal aspects separately.

### 3.2.3 EVALUATING LONG VIDEO GENERATION

With the emerging advancements, video generative models now aim to produce longer video sequences. Consequently, it becomes important to establish evaluation metrics capable of assessing long videos effectively. In Table 2, we demonstrate that STREAM is the only metric that provides accurate evaluations regardless of video length. Since FVD and VIS cannot be directly applied for long-video evaluation, we slightly modify these by measuring it for every 16 frames using a sliding

window, sFVD and sVIS. Here, we compare TATS-base, MoCoGAN-HD, DIGAN, and MeBT models. Our results show that, in the generation of 128 video frames, all these models present a notable decline in temporal naturalness compared to the generation of shorter, 16-frame video clips (Table 1). Notably, TATS-base demonstrates a significant drop in temporal coherence, which aligns well with the observations made in Yoo et al. (2023). This is readily apparent upon actual sampling (see Appendix A.8). In addition, by applying STREAM-S, it becomes evident that current video generative models are yet to overcome substantial challenges in generating realistic and diverse videos, particularly as the video length increases. These observations collectively highlight that there is still a long way to go for a natural, long video generation.

## 4 RELATED WORKS

**Video Generative Models**  Video generative models need to learn additional temporal information compared to image generation models. Considering this, various network architectures and training methods have been proposed. Tulyakov et al. (2018) and Li et al. (2019) have introduced a structure that combines a 2D convolutional network with an RNN to account for the temporal axis. Vondrick et al. (2016) have introduced a 3D convolutional structure that allows simultaneous consideration of spatial and temporal information. Another approach involves training a network to generate low-quality videos initially and progressively increasing the network size during training to enhance video quality gradually (Karras et al., 2017). Furthermore, networks based on two different architectures have been introduced to learn video content and motion separately, using one network to capture content and another to learn motion (Sun et al., 2020). All these prominent approaches rely on evaluation metrics to verify the effectiveness of networks in learning temporal information in videos. Therefore, in order to accurately assess the proposed methods, there is a need for new metrics that can consider the spatial and temporal quality separately.

**Video Evaluation Metrics**  In contrast to image generation, the video generation domain has only a limited number of metrics specifically designed for evaluation. Among these, Video Inception Score (VIS) and Fréchet Video Distance (FVD) are most commonly used. They both utilize the Inflated 3D Convnet (I3D) (Carreira & Zisserman, 2017). Given real image dataset $\mathcal{X} \in \mathbb{R}^d$, generated image dataset $g(\mathcal{Z}) \in \mathbb{R}^d$ through a generator, and set of labels $\mathcal{Y}$, IS computes the KL-divergence between the conditional real distribution $p(\mathcal{Y}|\mathcal{X})$ and marginal distribution $p(\mathcal{Y}) = \int_{z \in \mathcal{Z}} p(y \in \mathcal{Y}|g(z))dz$ by utilizing the softmax layer from the I3D network. On the other hand, FVD uses feature embeddings before the softmax layer of the I3D network, defining the real feature set $\mathcal{X} \in \mathbb{R}^d$ and the fake feature set $g(\mathcal{Z}) \in \mathbb{R}^d$, assuming that the distributions of these features are Gaussian. FVD calculates the Wasserstein distance between the given real distribution $\mathcal{X} \sim \mathcal{N}(\mu_\mathcal{X}, \Sigma_\mathcal{X})$ and fake distribution $g(\mathcal{Z}) \sim \mathcal{N}(\mu_{g(\mathcal{Z})}, \Sigma_{g(\mathcal{Z})})$ as follows: $FVD(\mathcal{X}, g(\mathcal{Z})) := |\mu_\mathcal{X} - \mu_{g(\mathcal{Z})}|^2 + Tr(\Sigma_\mathcal{X} + \Sigma_{g(\mathcal{Z})} - 2(\Sigma_\mathcal{X}\Sigma_{g(\mathcal{Z})})^{\frac{1}{2}})$. While these metrics offer initial insights, VIS and FVD each provide only a single, composite score, which limits their capability to elucidate the diverse strengths and weaknesses inherent in video generative models. This obscures the nuanced variances and subtleties in model performances, yielding a potentially superficial evaluation of the models' generative capacities. This limitation becomes especially significant in comparison to metrics like Precision and Recall, as proposed by Kynkäänniemi et al. (2019) for image generation tasks. These metrics, by separately evaluating the fidelity and diversity of generated image quality, allow for a more granular and nuanced understanding of model capabilities and shortcomings that is equally important in the domain of video generation.

## 5 CONCLUSION

Current evaluation metrics for video generative models simply extend the metrics originally designed for image generative models. Unlike images, however, video data requires careful consideration of temporal naturalness, which FVD fails to adequately address. In addition, the current state of performance measurement and analysis for video generative models is constrained by a limited set of metrics. To address this, we propose STREAM, a new video evaluation metric, which allows for the evaluation of realism, diversity, and temporal aspects of videos, independently. Through carefully designed experiments, we have shown that STREAM is an effective metric that can evaluate and analyze the performance of video generative models in a broader perspective.

ACKNOWLEDGMENTS

This work was supported by the National Research Foundation of Korea (NRF) grant funded by the Korea government (MSIT) (No.2022R1C1C1008496), Institute of Information & communications Technology Planning & Evaluation (IITP) grant funded by the Korea government (MSIT) (No.2020-0-01336, Artificial Intelligence Graduate School Program (UNIST), No.2021-0-02068, Artificial Intelligence Innovation Hub, No.2022-0-00959, (Part 2) Few-Shot Learning of Causal Inference in Vision and Language for Decision Making, No.2022-0-00264, Comprehensive Video Understanding and Generation with Knowledge-based Deep Logic Neural Network).

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

# A APPENDIX

## A.1 MOMENT GENERATING FUNCTION

Moment Generating Function is stated as follows:

**Definition A.1.** (Moment Generating Function) (Schervish & DeGroot, 2014). Given random variable X with probability density function f(X), the moment generating function of X,

$$M_X(t) = E[e^{tX}] = \int_X e^{tX} f(X) dx,$$

exists if there is an $h > 0$ such that for all $t$ in $-h < t < h$.

Taking the derivative of the MGF with respect to $t$ allows us to calculate arbitrary-order moments of $X$, such as the first moment which is the mean, and by taking the second derivative, the second moment which is the variance.

## A.2 STREAM-T METRIC BASED ON THE VIDEO EMBEDDING NETWORK

In this section, we validate whether the STREAM-T metric behaves as intended when using a video embedding network instead of an image embedding network. In this experiment, we have used the Masked Video Auto-encoder V2 (Wang et al., 2023) which is widely known to effectively capture temporal information of video input in their latent space. Since STREAM-T gives a score based on the video's temporal naturalness, we aim for STREAM-T to be robust to the visual distortions when there is sufficient temporal information in the video. On the other hand, in cases where the order of frames is shuffled as in local swaps or when different frames suddenly appear as in global swaps in terms of temporal aspects, we expect STREAM-T to gradually decrease in response based on the number of swaps. As in Figure 7, when visual distortion occurs, STREAM-T does not behave as intended. This can be attributed to the fact that video embedding mixes both visual and temporal information of the video into the latent space. Additionally, when local swap occurs, STREAM-T does not react sensitively, whereas it reacts significantly to global swaps. Through these results, we confirm once again that in a feature space where visual and temporal information is not accurately distinguished, STREAM-T does not operate as intended.

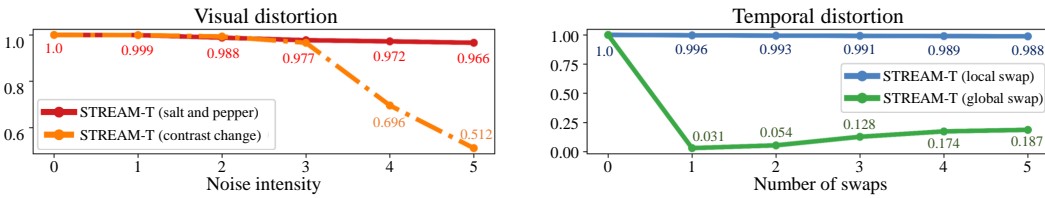

Figure 7: The behavior of STREAM-T when using a video embedding network instead of image embedding network. Visual distortion and temporal distortion experiments, based on UCF-101 dataset, are conducted using the same methods described in Section 3.1.1 and 3.1.2, respectively. In each experiment for visual distortion, the maximum noise intensity is applied while preserving the temporal flow of the video. The numbers within the figures represent the scores of the metrics.

## A.3 STREAM-T METRIC UTILIZING MEAN AMPLITUDE SIGNAL

When calculating STREAM-T, we do not use mean amplitude (i.e., $\hat{X}(\zeta_0 = 0)$), because it represents the average values of simple amplitude signals and does not provide additional information of temporal signal variations. In addition, the use of mean amplitude values would actually hinder STREAM-T from detecting the overall temporal signal changes. In Figure 8, STREAM-T, calculated including the mean amplitude, appears to be more sensitive to visual distortion compared to the previous result in Section 3.1.1. This indicates that mean amplitude leads the metric more dependent on visual information. Additionally, in Figure 8, STREAM-T becomes more sensitive to global swap when utilizing mean amplitude. In other words, considering the visual information provided by mean amplitude influences STREAM-T in a way that aligns with the trend of reduced diversity. Therefore, to accurately evaluate temporal naturalness with STREAM-T, it is better to exclude mean amplitude, which does not provide new information about temporal changes.

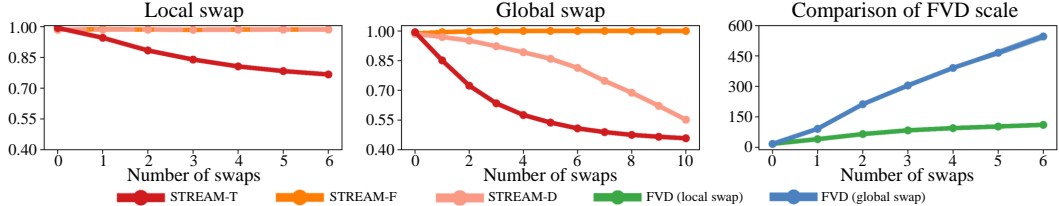

Figure 8: Behaviors of STREAM to various types of visual degradation when utilizing the mean amplitude signal. All the settings for this experiment is identical to the experiments in Figure 2.

Figure 9: Behaviors of STREAM to various types of temporal degradation when utilizing the mean amplitude signal. All the settings for this experiment is identical to the experiments in Figure 3.

## A.4 WHY STREAM-S UTILIZES AMPLITUDE AT FREQUENCY 0

In STREAM-S, we use the amplitude at frequency 0 (i.e., mean amplitude) calculations for two main reasons. First, the mean amplitude encapsulates the average signal information of a video, offering a comprehensive overview of its spatial characteristics. Second, including amplitudes across all frequencies renders P&R calculation inaccurate.

More specifically, P&R is designed to approximate the support of real and fake distributions, typically using the union of spheres centered on data points with radii equal to the distance to the $k$-nearest neighbor. These spheres are expected to cover at least $k$ data points (or image features) and collectively represent the population data support. However, videos, unlike images, are composed of multiple image frames. For instance, a 16-frame video yields 16 image features obtained by an embedding network. Applying FFT along the temporal axis, we get 8 amplitude features for each video. In this context, using P&R directly on videos might result in inaccurate data support estimation, as a single sphere may not even encompass all data points from one video. Naturally, in such a scenario, the approximated data support may fail to represent the full scope of video data. To address this, we use the mean amplitude, which represents the average feature of a video, as a single data point. This approach ensures a more accurate and representative approximation of real and fake video data support, enhancing the precision of STREAM-S measurements without limiting it to the number of video frames as shown in our experiments.

## A.5 TEMPORAL DISTORTION EXPERIMENT USING KINETICS-600 DATASET

Example of Kinetics-600 sample at random translation intensity 0

Example of Kinetics-600 sample at random translation intensity 3

Example of Kinetics-600 sample at random translation intensity 5

Example of Kinetics-600 sample at random translation intensity 7

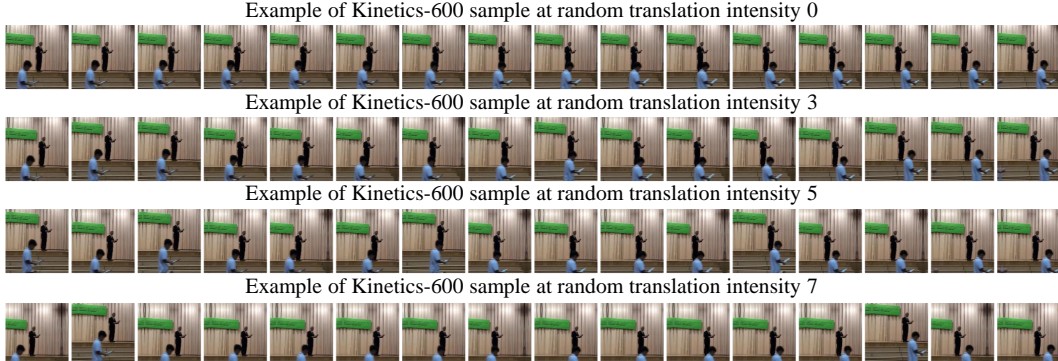

Figure 10: An example of Kinetics-600 data (in Figure 6 (a)) regarding the degree of distortion based on the random translation intensity. The information lost due to translation was compensated for through reflection. Similar to Section 3.1.3, this experiment employs translation based on intensity in random directions.

## A.6 SPEARMAN'S RANK CORRELATION COEFFICIENT WITH HUMAN JUDGEMENT SCORES

We have collected human judgement scores to quantify how well STREAM reflects human perceptual quality. We asked 81 raters to evaluate video generative models in Table 1 for each realism and temporal naturalness aspect. Each rater reviewed five randomly sampled videos from each model and assigned scores for realism and temporal naturalness. The scoring criteria for video realism is as follows: video scenes are indistinguishable from real video (6 points), scenes that are realistic and clearly interpretable (5 points), scenes are partially realistic (4 points), not realistic but recognizable scenes (3 points), only partially recognizable scenes (2 points), and complete inability to discern anything from the scenes (1 point). Additionally, the scoring criteria for temporal naturalness of videos is as follows: the scene transitions are smooth and continuous (3 points), some scene transitions are temporally inconsistent (2 points), and discontinuous scene transitions (1 point). Subsequently, we have measured the mean spearman's rank correlation coefficient (spearman's correlation) between model rankings based on human judgement scores for realism and temporal naturalness, and the model rankings from STREAM. From the results in Table 3, STREAM shows spearman's correlations of 0.9 and 0.6 for realism and temporal naturalness, respectively, confirming that it provides scores that align well with human perceptual quality. On the other hand, FVD has spearman's correlation of 0.7 and 0.5 for realism and temporal naturalness, respectively, indicating that it does not reflect human perceptual quality as effectively as STREAM. Note that, evaluating video diversity through human assessment is challenging to do fairly unless one reviews and remembers the features and types of a larger number of videos. Therefore, we do not proceed with additional experiments in this regard.

Table 3: Human evaluation results of video generative models listed in Table 1. Each spatial and temporal scrore denotes the sum of scores evaluated by 81 raters for each model, and the numbers in parentheses indicate the average scores for each model.

| Models | MoCoGAN-HD | DIGAN | TATS-base | VideoGPT | MeBT |
|---|---|---|---|---|---|
| Spatial score | 183 (2.25) | 237 (2.92) | 378 (4.66) | 304 (3.75) | 356 (4.39) |
| Temporal score | 122 (1.50) | 141 (1.74) | 161 (1.98) | 129 (1.59) | 145 (1.79) |

## A.7 Example of sample quality from short frame video generative models

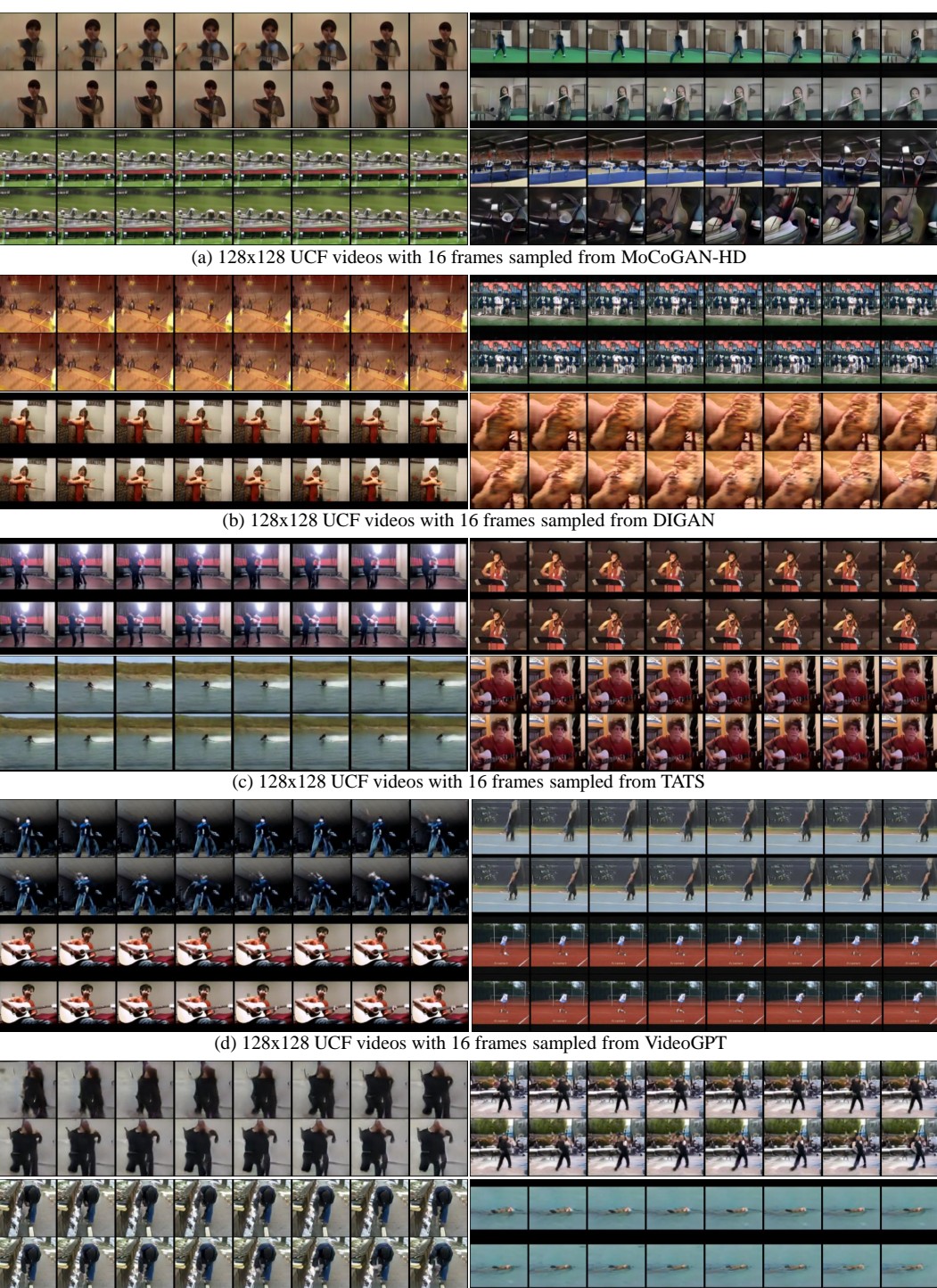

(a) 128x128 UCF videos with 16 frames sampled from MoCoGAN-HD

(b) 128x128 UCF videos with 16 frames sampled from DIGAN

(c) 128x128 UCF videos with 16 frames sampled from TATS

(d) 128x128 UCF videos with 16 frames sampled from VideoGPT

(e) 128x128 UCF videos with 16 frames sampled from MeBT

Figure 11: Example of sample quality sampled from video generative models trained on UCF-101 dataset. All the models listed in this figure are compared through STREAM and FVD in Table 1.

## A.8 LONG VIDEO SAMPLE QUALITY OF TATS-BASE

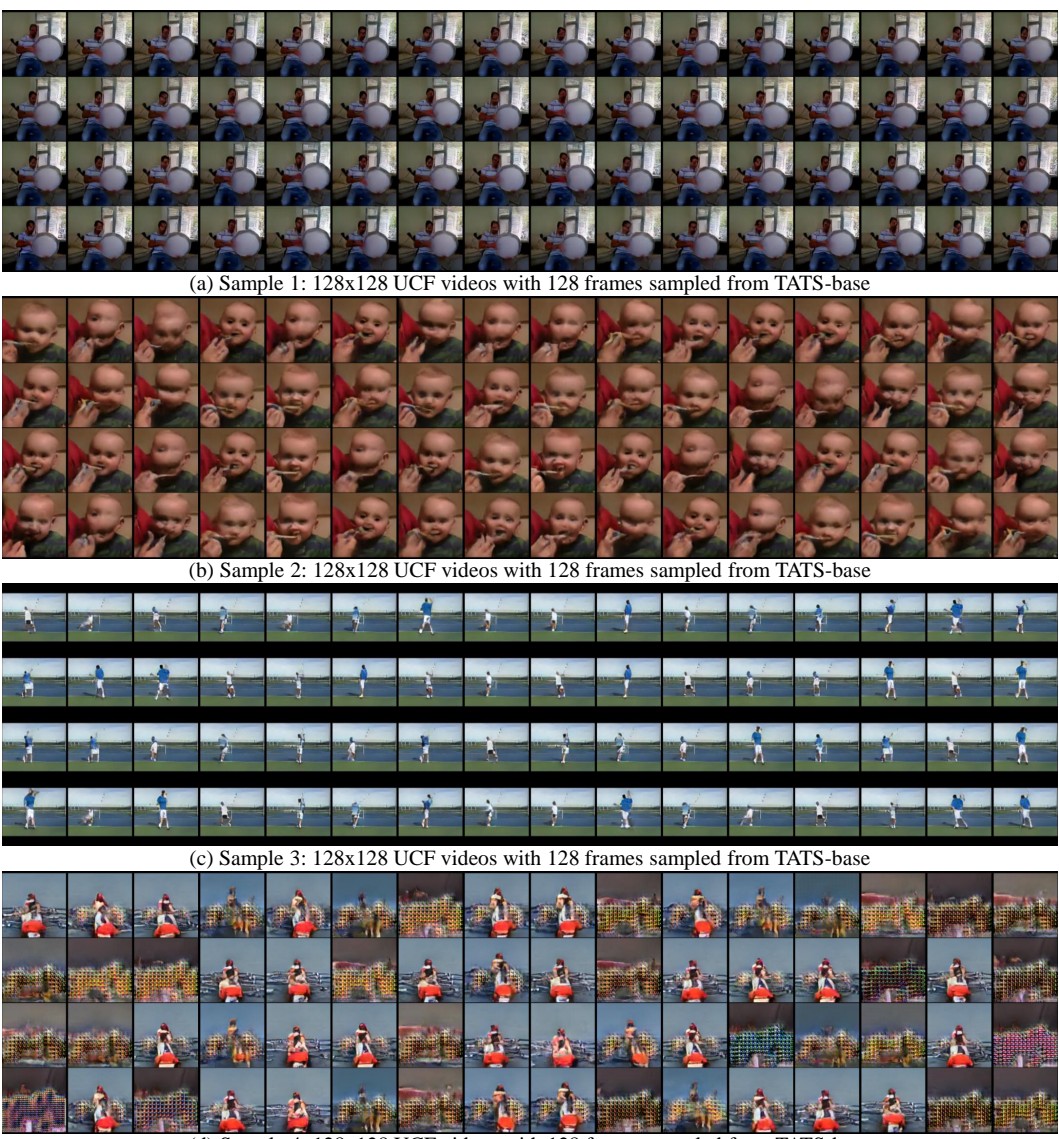

(a) Sample 1: 128x128 UCF videos with 128 frames sampled from TATS-base

(b) Sample 2: 128x128 UCF videos with 128 frames sampled from TATS-base

(c) Sample 3: 128x128 UCF videos with 128 frames sampled from TATS-base

(d) Sample 4: 128x128 UCF videos with 128 frames sampled from TATS-base

Figure 12: Example of sample quality of TATS-base trained on UCF-101 dataset. All listed video samples have 128 frames with a resolution of $128 \times 128$. We have visualized frames from the initial frame to the 64th frame for each video sample. The sample qualities from TATS-base resemble the stop scene (sample 1), local swap (sample 2), and global swap (sample 3, 4) cases in our experiments.

## A.9 EXAMPLE OF SAMPLE QUALITY FROM LONG FRAME VIDEO GENERATIVE MODELS

The figures in this section illustrate the sample quality of video generative models compared in Table 2. Examining the sample quality of MoCoGAN-HD, it is able to observe the model struggles to generate realistic frames but showing diverse frames with temporally continuous changes, which is consistent with the scores obtained from STREAM-F and T. In Figure 14, the frames sampled from DIGAN does not vary much over time and each frame does not have realism, corresponding to the low score of STREAM-F and T. The sample quality of MeBT, in Figure 15, shows relatively lower fidelity and temporal naturalness not having various changes over time, and this also aligns with the mediocre scores of STREAM-F and STREAM-T. By comparing the sample quality and STREAM score for each video generative model, STREAM seems to align with human perceptual quality.

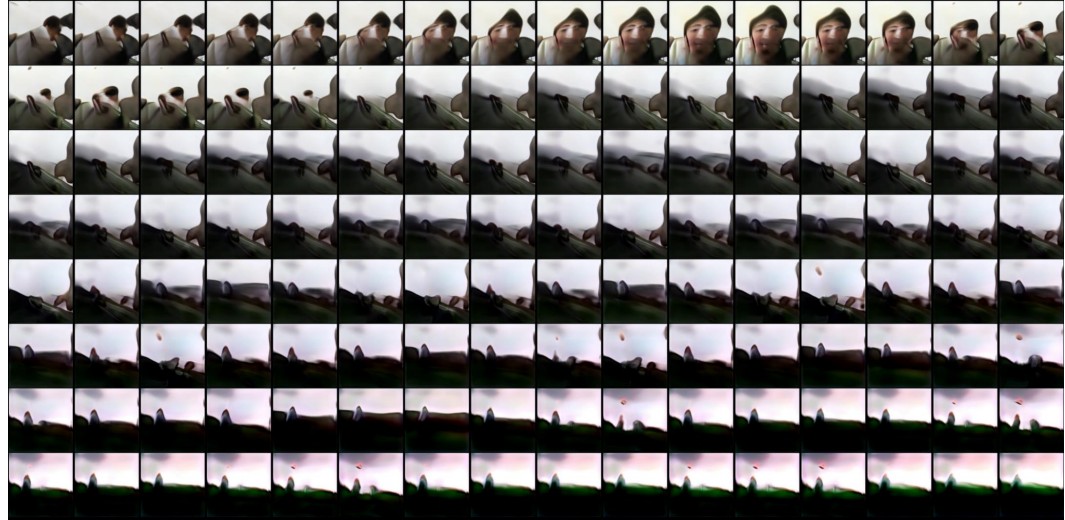

Figure 13: Example of sample quality of MoCoGAN-HD trained on UCF-101 dataset. The video has 128 frames with resolution of $\times 128 \times 128$.

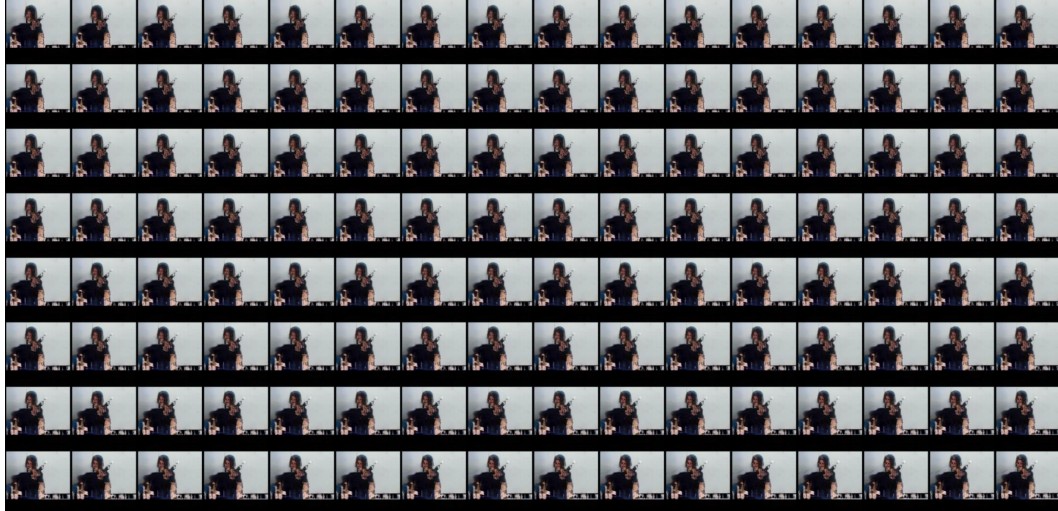

Figure 14: Example of sample quality of DIGAN trained on UCF-101 dataset. The video has 128 frames with resolution of $\times 128 \times 128$.

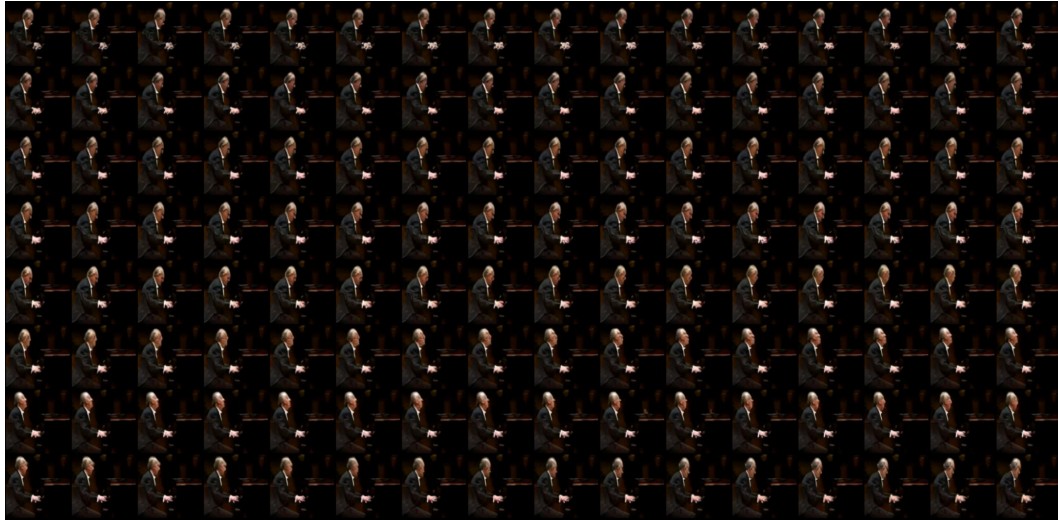

Figure 15: Example of sample quality of MeBT trained on UCF-101 dataset. The video has 128 frames with resolution of $\times 128 \times 128$.

## A.10 ABLATION STUDY ON THE DATA SIZE

Video data requires significantly larger memory capacity than images, highlighting the importance of evaluation metrics that can perform accurately with limited data size. We conducted an ablation study to determine at which data size FVD and STREAM-T are capable to provide consistent scores. For the experiment, we compared three types of distorted video data with real ones using local swap, global swap, and random translation. For each temporal distortions, the noise intensity is randomly chosen. As shown in Figure 16, STREAM-T shows consistent evaluation performance at approximately a data size of 2,000.

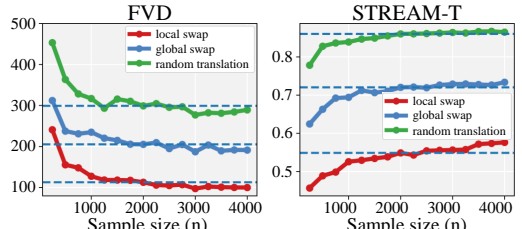

Figure 16: Ablation study on data size. The y-axis represents the scores of each metric, and the dashed lines indicate the metric values when data size is 2,000. The noise intensity of local swap, global swap, and random translation are randomly chosen.

## A.11 ABLATION STUDY ON THE HISTOGRAM BIN SIZE

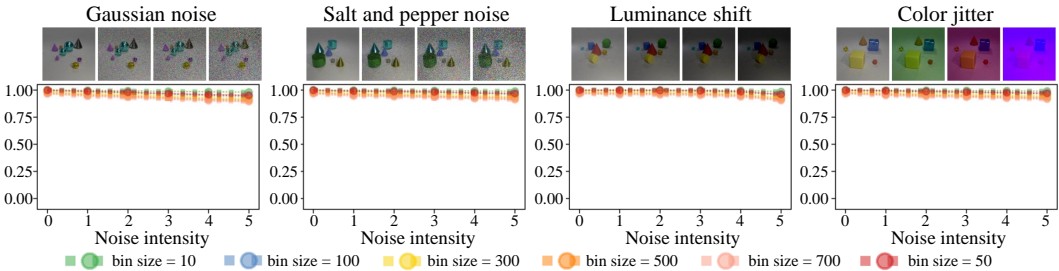

Figure 17: Robustness of STREAM-T against spatial noise across various bin sizes. Ablation study on bin size of STREAM-T. This investigation is built upon the same experimental setup as in Section 3.1.1. For each spatial noise scenario, we validate whether the evaluation tendencies change based on different settings of the bin size, which is a hyper-parameter of STREAM-T.

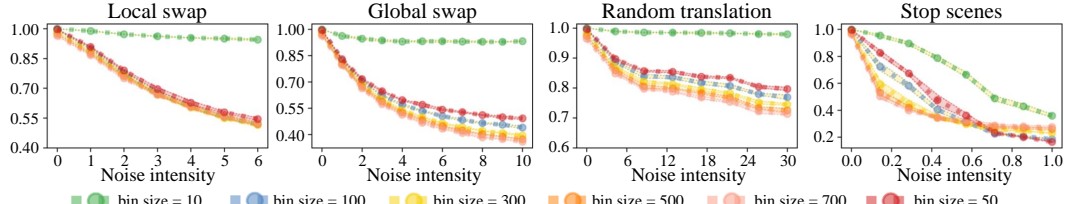

Figure 18: Consistency of STREAM-T against temporal distortions across various bin sizes. This investigation is built upon the same experimental setup as in Section 3.1.2 and 3.1.3. For each temporal distortion, we validate whether the evaluation tendencies change based on different settings of the bin size, which is a hyper-parameter of STREAM-T.

We have conducted an ablation study on histogram bin size which is a hyper-parameter of STREAM-T. In Figure 17, STREAM-T exhibits robustness across various types of visual noise regardless of bin size. Furthermore, in the presence of various types of temporal distortions (Figure 18), STREAM-T demonstrates consistent evaluations, except when the bin size becomes extremely small. We have selected the bin size ($= 50$) that exhibits the most optimal performance.

## A.12    EVALUATING VISUAL QUALITY DEGRADATION IN VIDEO WITH GAUSSIAN BLUR

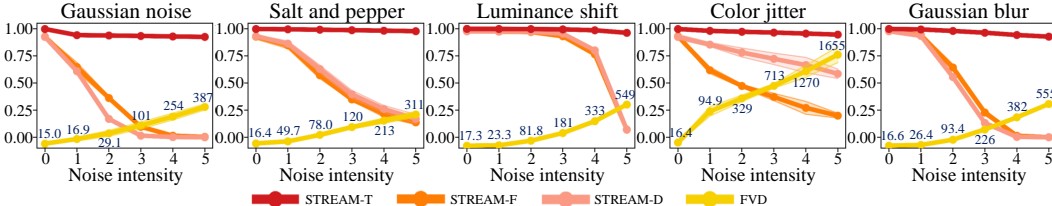

Figure 19: Comparison of metrics considering Gaussian blur distortion in addition to the experiment of Section 3.1.1. STREAM-T, which evaluates the temporal naturalness of videos, should demonstrate a robust evaluation trend for all types of visual distortions. We measure the extent of visual distortion and decrement in video diversity through STREAM-F and STREAM-D, respectively.

Video generative models often produce blurry samples, necessitating development of effective evaluation metrics capable of appropriately penalizing the visual quality of such samples. In Figure 19, we have observed that as the intensity of Gaussian blur increases, STREAM-S exhibits heightened sensitivity, ultimately converging to a score of 0. This responsiveness underscores STREAM-S's ability to discern and penalize the degradation in visual quality associated with intensifying blur.

## A.13    VALIDATION OF STREAM-T BY COMPARING FAST AND SLOW VIDEO DATASET

Table 4: Experiment on the evaluation trends of STREAM-T with variations in Video Frame Per Second (FPS). Real video exhibit a fast change in frame rate at $\times 1.0$ FPS, while setting the fake video to have progressively decreasing speed of change from $\times 1.0$ FPS to $\times 0.2$ FPS.

| Fake FPS | $\times 1.0$ FPS | $\times 0.8$ FPS | $\times 0.6$ FPS | $\times 0.4$ FPS | $\times 0.2$ FPS |
|---|---|---|---|---|---|
| STREAM-T | 0.996 ($\pm$0.000) | 0.765 ($\pm$0.003) | 0.764 ($\pm$0.003) | 0.418 ($\pm$0.003) | 0.352 ($\pm$0.004) |

The objective of video generative models is to resemble the spatial and temporal characteristics of real dataset by estimating the real data distribution. Therefore, when there are differences in movements of real and fake video datasets, the video generative model has estimated the real distribution with a temporal difference. In Table 4, we have compared the real videos with fast FPS and generated videos with slow FPS using STREAM-T. STREAM-T has a decreased value when the generated video has a relatively low FPS compared to the real video, which shows the effectiveness of STREAM-T in capturing the temporal difference.

## A.14 Additional synthetic experiment with using UCF-101 dataset

Table 5: Study on the stability of STREAM metric. To verify the stability of STREAM metric in a few corner cases, we consider two experimental settings, reverse and flip. The first experiment, 'reverse', involves reversing the order of video frames in each video sample. The 'flip' experiment verifies the realism of video dataset created by horizontally flipping all frames of each video. Lastly, the 'reverse & flip' applies both the 'reverse' and 'flip' settings simultaneously.

|  | VIS ($\uparrow$) | FVD ($\downarrow$) | STREAM-T ($\uparrow$) | STREAM-F ($\uparrow$) | STREAM-D ($\uparrow$) |
|---|---|---|---|---|---|
| Reference | 79.0769 | 14.3820 | 0.9989 | 0.9833 | 0.9842 |
| Reverse | 78.3968 | 24.9380 | 0.9989 | 0.9854 | 0.9825 |
| Flip | 78.5281 | 36.0726 | 0.9954 | 0.9843 | 0.9812 |
| Reverse & Flip | 78.5338 | 40.0170 | 0.9954 | 0.9827 | 0.9827 |

Table 6: Comparison of evaluation trends between STREAM-T and FVD when the order of partial video sequence is reversed. This experiment involves selecting three consecutive frames randomly from 16 frames of a real video and reversing the order of these three frames when the reverse intensity is set to 1. As the value of the reverse intensity increases, the process described for intensity 1 is repeated for the specified number of times (i.e., reverse intensity). An ideal metric should give progressively worse scores with increasing reverse intensity.

| Reverse intensity | 0 | 2 | 4 | 6 | 8 | 10 | 12 |
|---|---|---|---|---|---|---|---|
| STREAM-T | 0.9948 | 0.9688 | 0.9262 | 0.8865 | 0.8451 | 0.8231 | 0.7916 |
| FVD | 74.203 | 85.229 | 83.641 | 93.113 | 98.363 | 115.98 | 113.09 |

Given the nature of our metric, which focuses on evaluating continuity and consistency through frequency analysis, STREAM-T remains agnostic to the direction of time. However, this unique characteristic allows STREAM-T to be sensitive not only to situations where only a portion of the video sequence is reversed but also to various types of temporal inconsistencies. However, it is important to note that the generation of a completely reversed video is an uncommon scenario, and in such cases all the metrics (STREAM-T, VIS, and FVD) may not provide optimal responses. Still, other than this rare case, STREAM demonstrates its effectiveness in detecting partially reversed segments within videos that are otherwise temporally natural (Table 6). For example, when the video is entirely reversed on the temporal axis, FVD shows a modest increase from 14.38 to 24.93. This change, while noticeable, is relatively small compared to significant fluctuations (such as those in the hundreds) observed in other cases (Figure 5 and 6 in Section 3.2.1). Thus, in practice, this minor variation would make it difficult to conclusively determine if FVD is significantly impacted by this aspect alone.

The spatially flipped videos, being the mirror image of each video frame, possess highly realistic spatial and temporal quality. Through the experiment, we have observed that STREAM and VIS remains unchanged, while FVD slightly increased from 14.38 to 36.07. When both reverse and flipping is applied, the metrics should provide scores indicating spatially and temporally realistic results. From the results, VIS and STREAM consistently exhibited scores identical to the reference score, whereas FVD showed an increased value from the reference score (14.38) to 40.01.

## A.15 SENSITIVENESS OF VIS TOWARD SPATIAL AND TEMPORAL DEGRADATIONS

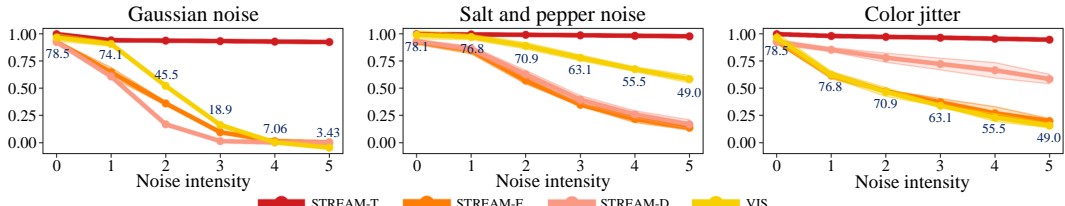

Figure 20: Comparison of VIS and STREAM under various types of visual degradation on UCF-101 data. Gaussian noise, salt and pepper noise, and color jitter are conducted on the same experimental setup as Section 3.2.1.

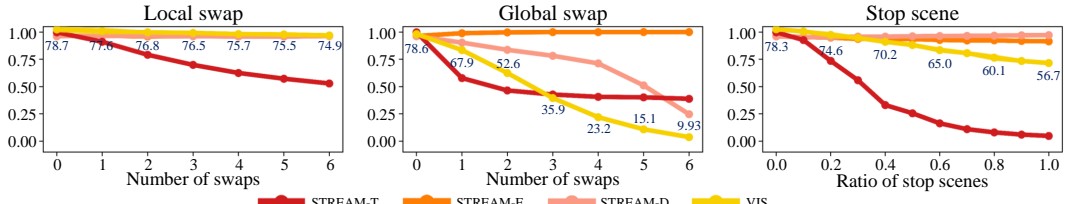

Figure 21: Comparison of VIS and STREAM under various types of temporal degradation on UCF-101 data, aiming to determine whether VIS respones to changes in temporal. Local and global swap conducted on the same experimental setup as Section 3.1.2 and stop scenes conducted under same conditions as

3.1.3.

We conducted experiments in Section 3.2.1 using the UCF-101 dataset to compare STREAM and VIS when spatial distortions such as gaussian noise, salt and pepper noise, and color jitter are applied to videos. Similarly, for temporal distortion, we performed experiments using local swap, global swap, and stop scene in Section 3.1.2 and 3.1.3 to compare the evaluation trends of STREAM-S and VIS. In Figure 20, both STREAM and VIS react to spatial distortion. However, for temporal distortion, as shown in Figure 21, VIS cannot discern different levels of temporal distortions compared to STREAM-T.

## A.16 ADDITIONAL SYNTHETIC EXPERIMENT WITH USING TAICHI DATASET

Table 7: Comparison and analysis of video generative models (unconditional). All the models are trained on the Taichi dataset (Siarohin et al., 2019) generating 16 frame videos. The numbers in parentheses next to evaluation scores represent the standard deviation of the scores, calculated through ten repeated measurements. Note that LVDM generates videos at a resolution of 256x256; for comparison purposes, we have downscaled them to 128x128 resolution.

|  | Resolution | FVD ($\downarrow$) | STREAM-T ($\uparrow$) | STREAM-F ($\uparrow$) | STREAM-D ($\uparrow$) |
|---|---|---|---|---|---|
| DIGAN | 128×128 | 728.7 ($\pm$12.21) | 0.7711 ($\pm$0.0050) | 0.7258 ($\pm$0.0693) | 0.1901 ($\pm$0.0152) |
| TATS | 128×128 | 445.6 ($\pm$13.63) | 0.8712 ($\pm$0.0045) | 0.7197 ($\pm$0.0882) | 0.4024 ($\pm$0.0244) |
| MeBT | 128×128 | 441.7 ($\pm$11.69) | 0.8561 ($\pm$0.0033) | 0.6712 ($\pm$0.0672) | 0.3557 ($\pm$0.0186) |
| LVDM | 128×128 | 205.5 ($\pm$6.619) | 0.9356 ($\pm$0.0017) | 0.8291 ($\pm$0.0575) | 0.4829 ($\pm$0.0288) |

## A.17 THE EFFECT OF FRAME RESOLUTION AND VIDEO LENGTH ON STREAM

In this section, we have conducted experiments to assess STREAM's performance across different video resolutions and lengths under visual or temporal distortion. We have used gaussian blur (for visual distortion) and local swap (for temporal distortion).

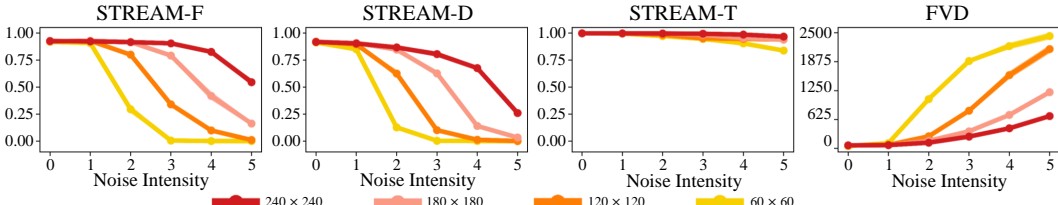

Figure 22: Performance of STREAM for the visual distortions on 16 frame videos with "various resolutions". In this experiment, we increase the strength of gaussian blur noise to verify whether STREAM-S exhibits a gradual decrease in scores, while STREAM-T maintains consistent scores. Since the intensity of noise applied to the video is same at each noise level, important information in videos with lower resolutions is destroyed more rapidly compared to higher resolutions. Therefore, STREAM should react more sensitively as videos have lower resolutions.

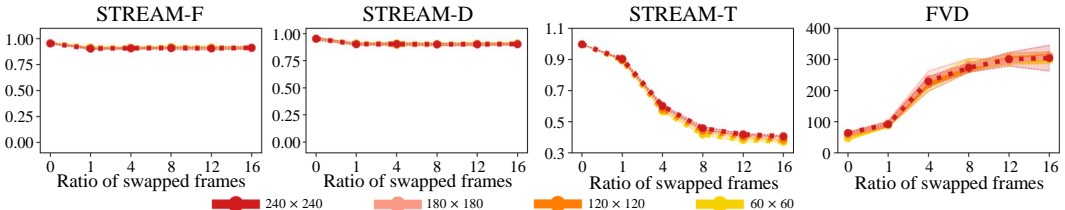

Figure 23: Performance of STREAM for the temporal distortion on 16 frame videos with "various resolutions". In this experiment, using the same setup as the local swap in Section 3.1.2, we verify whether STREAM-T appropriately responds to temporal distortion as the number of swaps increase.

We tested video resolutions of 60×60, 120×120, 180×180, and 240×240. Applying a uniform gaussian blur across resolutions (in Figure 22), we noted a more pronounced degradation in lower resolution videos, where fewer pixels carry critical information. This trend is accurately captured by the metrics. In the local swap tests (in Figure 23), we shuffled video frames equally across resolutions. Since local swap affects only temporal quality without impacting the spatial quality of the video, an ideal metric should consistently provide low temporal scores as the number of local swaps increases, regardless of resolution. Our results show that FVD and STREAM meet these criteria.

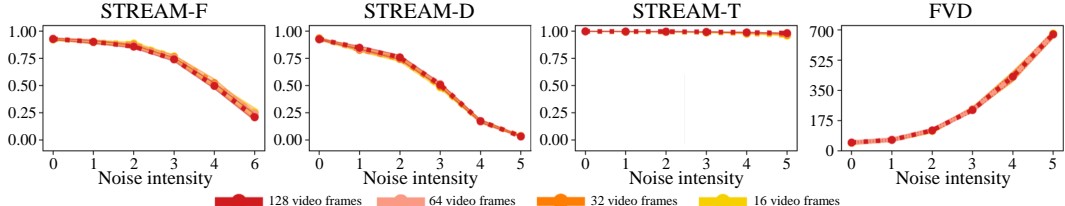

Figure 24: Performance of STREAM for the visual distortions on videos with "various video length". In this experiment, we increase the strength of gaussian blur noise to verify whether STREAM-S exhibits a gradual decrease in scores, while STREAM-T maintains consistent scores. Note that, FVD is measured for every 16 frames using a sliding window with stride of 16.

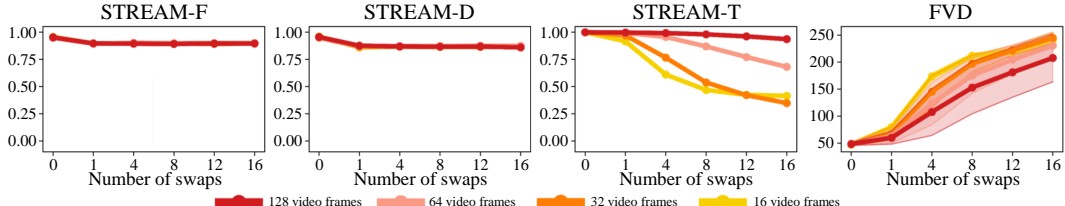

Figure 25: Performance of STREAM for the temporal distortion on videos with "various video length". In this experiment, using the same setup as the local swap in Section 3.1.2, we verify whether STREAM-T appropriately responds to temporal distortion as the number of swaps increase. Given that local swapping is uniformly applied to videos irrespective of video length, shorter videos exhibit greater temporal degradation when swapping occurs compared to longer videos. Therefore, ideal temporal metrics should adeptly capture the extent of temporal distortion with respect to video length. Note that, FVD is measured for every 16 frames using a sliding window with stride of 16.

We tested video lengths of 16, 32, 64, and 128 frames. When applying a uniform gaussian blur (in Figure 24), the spatial quality reduction was consistent across lengths, without temporal quality degradation. Both STREAM and FVD effectively captured this behavior. For local swaps, we used a fixed number of swaps across different lengths (in Figure 25). Consequently, local swaps affect longer videos less, as the proportion of unshuffled frames increases. STREAM demonstrated better temporal consistency for longer videos, whereas FVD struggled to reflect this variation. Additionally, FVD exhibited increasing deviations in evaluation scores as the video length increases. This highlights FVD's limitations in evaluating videos longer than 16 frames, as it tends to over or underestimate temporal distortions. Our findings show the credibility of STREAM in handling various video resolutions and lengths, and they provide valuable insights into the limitations of existing metrics in such contexts.

## A.18    EVALUATION OF VIDEO PREDICTION MODELS USING STREAM

Table 8: Comparison and analysis of video prediction models. All the models are trained on the BAIR dataset predicting 15 frame vidoes with $64 \times 64$ resolution from 1 conditional frame. The numbers in parentheses next to evaluation scores represent the standard deviation of the scores, calculated through five repeated measurements.

|        | FVD ($\downarrow$) | STREAM-T ($\uparrow$) | STREAM-F ($\uparrow$) | STREAM-D ($\uparrow$) |
|--------|--------------------|-----------------------|-----------------------|-----------------------|
| RaMViD | 160.1 ($\pm$2.719) | 0.9193 ($\pm$0.0007)  | 0.5562 ($\pm$0.0151)  | 0.4709 ($\pm$0.0161)  |
| MCVD   | 21.63 ($\pm$0.511) | 0.9896 ($\pm$0.0002)  | 0.8355 ($\pm$0.0076)  | 0.8885 ($\pm$0.0086)  |

In this section, we demonstrate that STREAM is not limited to solely comparing unconditional video generative models; rather, it is equally applicable to a broader range of tasks, including video prediction models. The objective of video prediction model is to learn representations of objects and their temporal changes from given past frames and predict the future video scenes. Therefore, we utilize the dataset consisting of input-output pairs of video prediction models as real and fake

datasets for evaluation. We have compared open-source models, RaMViD(Höppe et al., 2022) and MCVD(Voleti et al., 2022), trained on BAIR(Ebert et al., 2017) dataset predicting 16 frame video with $64 \times 64$ resolution.

In Table 8, the FVD score indicate MCVD's superiority in generating better predictions than RaMViD. STREAM offers a detailed analysis of each model's performance in terms of fidelity, diversity, and temporal naturalness. While MCVD excels in all aspects, RaMViD lags, particularly in fidelity and diversity. In our detailed assessment (in Figure 27 and 26), we observe that RaMViD tends to produce videos with blurry scenes containing artifacts, which are the source of its low fidelity score. Moreover, it shows temporally inconsistent transitions; e.g., objects in the video tend to suddenly appear or disappear over time. On the other hand, MCVD's outputs are of higher fidelity and demonstrate smoother, more consistent transitions with fewer discrepancies in object appearance during scene transitions.

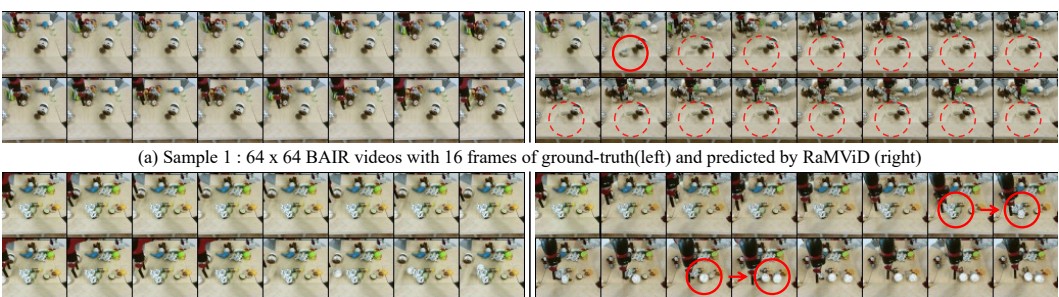

(a) Sample 1 : 64 x 64 BAIR videos with 16 frames of ground-truth(left) and predicted by RaMViD (right)

(b) Sample 2 : 64 x 64 BAIR videos with 16 frames of ground-truth(left) and predicted by RaMViD (right)

Figure 26: Example of samples predicted by the RaMViD model: predicted result (right), ground-truth (left). The red circle in (a) demonstrates the emergence of blurry artifacts. The red circle in (b) illustrates instances where the form of the object undergoes transformation over times and sudden appearances.

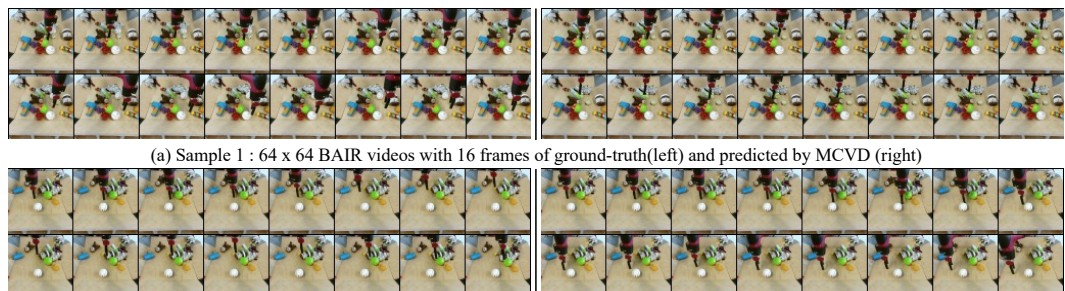

(a) Sample 1 : 64 x 64 BAIR videos with 16 frames of ground-truth(left) and predicted by MCVD (right)

(b) Sample 2 : 64 x 64 BAIR videos with 16 frames of ground-truth(left) and predicted by MCVD (right)

Figure 27: Example of samples predicted by the MCVD model: predicted result (right), ground-truth (left). Through example, predicted samples exhibit similar spatial and temporal quality compared to real video. Additionally, the occurrence of blurry artifacts or sudden appearances and disappearances of objects in each video scene is less frequent compared to the sample quality of RaMViD

