# OpenReview forum: "STREAM: Spatio-TempoRal Evaluation and  Analysis Metric for Video Generative Models"
_ICLR.cc/2024/Conference — ICLR 2024 poster_

### Official Review · Reviewer_srFK · 2023-10-23

**Soundness:** 3 good
**Presentation:** 3 good
**Contribution:** 2 fair
**Rating:** 6
**Confidence:** 4

**Summary:**

The authors propose a set of metrics to evaluate generative models for video. The propose STREAM-T to assess temporal naturalness as well as STREAM-F for fidelity of videos and STREAM-D for the diversity of videos. They compare to the Frechet Video Distance (FVD) on both synthetic data and using a number of generative models.

STREAM-T is based on the idea of looking at FFT features of the video to assess temporal "naturalness", while STREAM-F and STREAM-D are based on Precision & Recall metrics, but include tweaks to make them work better for video data.

**Strengths:**

**Originality:** The Authors argue that STREAM can separately assess temporal and spatial aspects of video, and works regardless of video length (unlike FVD). To the best of my knowledge this is true. I like their idea of using FFT to look at spectral features to identify temporal consistency.

**Quality:** The motivation for their method makes sense, and I think their empirical valuation is sensible. The authors only compare to FVD.

**Clarity:** I was able to follow along nicely

**Significance:** I think it is certainly nice to be able to have several dimensions upon which to evaluate generative methods.  However, it is unclear to me how useful that is in practice. Having these metrics can certainly help "debug" generative methods, but I would imagine that they will not be as useful as a "one metric to judge overall quality" that FVD provides. Especially since the human evaluation shows that FVD actually does fairly well (especially given that in contrast to the three measurements of STREAM it  is only a single number).


**UPDATE AFTER READING THE REBUTTAL**
Overall, my judgment is that the authors have convinced me that in principle this manuscript deserves publication: they attack a meaningful problem and their empirical work is solid. So I think this work scores high enough in Originality,  Quality, and Clarity to be publication. As far as signficance goes: I'm not 100% conviced ICLR is the best venue for this work (I'd imagine you find a more interested audience in conferences that are more focused on computer vision), which is why I still think the manuscript is only marginally above the acceptance threshold for ICLR.

**Weaknesses:**

* While I think it is useful to have metrics that focus on different aspects of the generation, I would imagine that while developing a method, most of the time it is more helpful to have a single metric to look at to judge progress (if I'm developing a new method and my newest change to the method improves STREAM-F but hurts STREAM-T, is it a good modification or not. Thus, I think the proposed STREAM metrics will be helpful, but will fail to actually replace FVD.  If the authors could find a good way to combine their measurements into a single number (e.g. akin to the F1-score to combine P&R), I think the paper would have more impact.
* The paper mentions the Video Inception Score (VIS) several times as go-to metric for this task, but does not compare to it. The authors should motivate why they did not use this metric at all, despite it being an obvious competitor.
* I find it confusing that the authors introduce the term "STREAM-S": Its name implies that it's one of the metrics that is being introduced (it follows the same naming convention), but it actually isn't. I'm absolutely unclear what that term actually denots.. I think the paper would improve in clarity if that term would be removed and instead clearly state that there are 3 new metrics that are being introduced (STREAM-T, STREAM-F, STREAM-D).

## Typos and Minor points
Page two: Spatio-TemproRal Evaluation and Analysis Metric(STREAM)  => TempoRal
Section 2.1: "Let real and fake video datasets as"  => ... be denoted as...
Seciton 2.2: we utilize SwAV (Caron et al., 2020), a proficient image embedding network => SwAV is an algorithm, not a network

"In the image generation task, Kynka¨anniemi et al. ¨ (2019) have proposed precision and recall (P&R) to separately evaluate the fidelity and diversity of the generated image quality." => I think this was first proposed by Sajjadi et al, (NeurIPS 2018), Assessing Generative Models via Precision and Recall.

Section 2.4: "If we define a sphere with centered at sample point"  => I don't understand this sentence

**Questions:**

* I would like to know how well FVD correlates with the various STREAM measures. Figures 3, 4 and 6 seem to indicate that the spearman correlation between STREAM-T and FVD is actually fairly high, which is completely contrary to the main text, which several times claims that FVD is not good at picking up temporal details.

* What is the reason for not comparing against VIS?

* STREAM-T uses a histogram comparison. How do binning sizes affect the outcome?

* Will the authors provide source code for implementing STREAM? It seems nontrivial to implement.

---

> ### Author Response · Authors · 2023-11-18
>
> # Reviewer 4
> **R4-W1. Find a good way to combine their measurements into a single number (e.g., F1-score):**
> Please see our GC2 in general comments.
> &ensp;
>
> **R4-W2 & Q2. Compare with Video Inception Score (VIS) (Table 1, 2, 5, Figure 20 and 21):**
> We have added VIS for all the experiments based on the UCF-101 dataset. Additionally, we have compared STREAM with VIS under various temporal distortion scenarios in Figure 21 (Appendix A.15). The experimental result shows that VIS cannot discern different levels of temporal distortions such as shuffling video frames within the video or replacement of video with static scenes (which are identical settings to Figure 3 and 4).
> &ensp;
>
> **R4-W3. Clearly state that there are 3 new metrics (STREAM-T, F, and D):** STREAM-S stands for STREAM-Spatial quality, composed of STREAM-F and STREAM-D. As suggested by the reviewer, we will make it clearer that we are introducing 3 new metrics (STREAM-T, F, and D).
> &ensp;
>
> **R4-Q1. Spearman’s correlation between STREAM-T and FVD is fairly high, which is completely contrary to the main text:**
> We appreciate the reviewer’s inquiry about the correlation between FVD and STREAM measures. It is important to clarify that our main argument was not on asserting that FVD is ineffective at detecting temporal details, but rather on FVD has a relative bias towards spatial quality over temporal quality. This distinction is crucial. While Figures 3, 4, and 6 do show a fair correlation between STREAM and FVD, this correlation primarily reflects the fact that both metrics are responsive to changes in video quality, albeit with different emphasis. STREAM is designed to more finely discern both spatial and temporal aspects, whereas FVD, despite its ability to detect temporal changes, tends to be more influenced by spatial quality. Therefore, the observed correlation does not necessarily contradict our main argument but rather highlights the nuanced ways in which these metrics assess video quality.
>
> In fact, it is natural to have a fairly high correlation between STREAM and FVD as both scores have a linearly increasing or decreasing trend as the intensity of temporal degradation increases. However, in this context, as FVD computes a single value for spatio-temporal quality, it remains unclear whether FVD genuinely responds to temporal distortion or exhibits decreasing trend in response to spatial aspects.
> &ensp;
>
> **R4-Q3. How do bin sizes affect the outcome (Figure 17 and 18):**
> STREAM-T exhibits robustness across various types of visual noise regardless of bin size. We have added an ablation study on histogram bin size in **Figure 17 and 18 (in Appendix A.11)**. Furthermore, in the presence of various types of temporal distortions, STREAM-T demonstrates consistent evaluations, except when the bin size becomes extremely small. We have selected the bin size ($= 50$) that exhibits the most optimal performance.
> &ensp;
>
> **R4-Q4. Provide source code for implementing STREAM:**
> We have already provided the link of our code at the end of Abstract. Additionally, we are planning to add an example page that demonstrates how the current implementation can be easily and simply executed with just a few lines of code.
> &ensp;
>
> ***Minor suggestions)**
> **(1) “Let real and fake video datasets as X and Y=> be denoted as”:**
> We have revised this part as ”Let real and fake video datasets be denoted as X and Y, respectively”.
>
> **(2) “SwAV is an algorithm, not a network”:**
> As advised by the reviewer, we have modified this part to “we utilize a proficient image embedding network proposed by Caron et al. (2020)”.
>
> **(3) “P\&R is first proposed by Sajjadi et al, (NeurIPS 2018)”:**
> We have clarified this part by explicitly changing it to “improved precision and recall (P\&R)”.
>
> **(4) “If we define a sphere with centered at sample point => I don’t understand this”:**
> This part explains the issues encountered when applying the P\&R support estimation method straight-forwardly. For better clarity, we explicitly mentioned in the main text as “In the process of P\&R support estimation” and provided a more detailed discussion in Appendix A.4.

---

> > ### Comment · Reviewer_srFK · 2023-11-22
> >
> > Thank you for your detailed rebuttal. It adresses my main concerns (I still think removing the concept of STREAM-S would improve readability of the manuscript, but that's just a subjective opinion), and I will adjust my review score accordingly.
> >
> > Overall, my judgment is that the authors have convinced me that in principle this manuscript deserves publication: they attack a meaningful problem and their empirical work is solid. So I think this work scores high enough in Originality,  Quality, and Clarity to be publication. As far as signficance goes: I'm not 100% conviced ICLR is the best venue for this work (I'd imagine you find a more interested audience in conferences that are more focused on computer vision), which is why I won't increase my score beyond a "borderline accept".

---

### Official Review · Reviewer_rxEA · 2023-10-26

**Soundness:** 2 fair
**Presentation:** 3 good
**Contribution:** 2 fair
**Rating:** 3
**Confidence:** 5

**Summary:**

The paper presents a new metric to evaluate the generated videos. The authors present three new metrics to assess video quality namely 1) spatial fidelity 2) diversity and 3) temporal coherence. Authors construct various kind of perturbation and evaluate the videos.

**Strengths:**

* Paper is easy to follow
* The most important contribution of the paper is the newly proposed metric is bounded between 0-1 as opposed to FVD, which is an unbounded metric.

**Weaknesses:**

* FVD is a single metric used to evaluate a video. With STREAM as a metric, you would have 3 sub-metrics to evaluate generated videos. This would result in $2^3$ scenarios when comparing two baselines, making it tedious to evaluate a new method.

**Questions:**

I would like to see how this metric performs in three scenarios.
* When the generated video consists of only one frame repeating throughout the video segment
* One of the main selling points of the FVD metric was it penalizes blur phenomena significantly higher than noise phenomena. This was a useful property because it correlates well with human vision. Additionally, the video generation methods tend to produce blurry samples which would score higher on traditional metrics like SSIM and PSNR. I would want to see how it performs on blurry videos (apply Gaussian blur).
* What would be the results if the video is reversed temporally and evaluated? please do the same for flipping the videos spatially(take mirror images of the frames) and lastly, run the evaluation for both spatially and temporally flipped video sequences.
I would like to see an evaluation of all these three scenarios before making my final decision.

How is the metric affected by the length and resolution of videos, and if it is affected, please provide the standardization of metric because people can game the metric utilizing this loophole.

---

> ### Author Response · Authors · 2023-11-18
>
> # Reviewer 3
> **R3-W1. It is tedious to evaluate a new method with 3 sub-metrics:**
> Please see our GC2 in general comments.
> &ensp;
>
> **R3-Q1. Experiment when video consists of only one frame repeating throughout the video segment (Figure 4 and 6):**
> Please see our GC1 in general comments.
> &ensp;
>
> **R3-Q2. Check STREAM metric with gaussian blur (Figure 19):**
> Thanks for pointing this out. Similar to the usefulness of FVD in practice, We have observed STREAM-S shows significant sensitivity toward Gaussian blur phenomena. For detailed experimental results, please refer to our **Figure 19 (in Appendix A.12).**
> &ensp;
>
> **R3-Q3.** We have conducted the experiments mentioned by the reviewer in **Table 5 (in Appendix A.14)**. Thank you for your comments. Your feedback greatly improved the understanding of our work. We will divide the reviewer’s question into three parts for clarity:
>
> **(1) Evaluation on temporally reversed videos:**
> First, it is indeed a valid point that STREAM-T does not assign low values when the frames of a video are completely reversed. Given the nature of our metric, which focuses on evaluating continuity and consistency through frequency analysis, STREAM-T remains agnostic to the direction of time. However, this unique characteristic allows STREAM-T to be sensitive not only to situations where only a portion of the video sequence is reversed but also to various types of temporal inconsistencies.
>
> However, it is important to note that the generation of a completely reversed video  is an uncommon scenario, and in such cases all the metrics (STREAM-T, VIS, and FVD) may not provide optimal responses.  Still, other than this rare case, STREAM demonstrates its effectiveness in detecting partially reversed segments within videos that are otherwise temporally natural (Table 6 in Appendix A.14). For example, when the video is entirely reversed on the  temporal axis, FVD shows a modest increase from 14.38 to 24.93. This change, while noticeable, is relatively small compared to significant fluctuations (such as those in the hundreds) observed in other cases (Figure 5 and 6 in Section  3.2.1). Thus, in practice, this minor variation would make it difficult to conclusively determine if FVD is significantly impacted by this aspect alone.
>
> Lastly, typical video data, including datasets like CATER, UCF-101, Kinetics, Taichi, and BAIR Robot Pushing, often features repetitive scenes, such as rotations, repetitive exercises, and cooking. In such practical scenarios, STREAM-T is effective in evaluating inconsistencies in continuity and consistency of temporal movements. This aspect, acknowledging the limitations and strengths of STREAM-T in handling various temporal dynamics, will be incorporated into the limitations section.
>
> **(2) Evaluation on spatially flipped videos:**
> Spatially flipped videos, being the mirror image of each video frame, possess highly realistic spatial and temporal quality. Through the experiment, we have observed that STREAM and VIS remains unchanged, while FVD slightly increased from 14.38 to 36.07.
>
> **(3) Evaluation on temporally reversed and spatially flipped videos:**
> This experiment simply applied the above two cases simultaneously, and the metrics should provide scores indicating spatially and temporally realistic results. From the results, VIS and STREAM consistently exhibited scores identical to the reference score, whereas FVD showed an increased value from the reference score (14.38) to 40.01.

---

> ### Author Response · Authors · 2023-11-18
>
> **R3-Q4.** For clarity, we will address the reviewer’s question by separately explaining the aspects related to resolution and length.
>
> **(1) How is the metric affected by the resolution of videos:**
> STREAM, along with the majority of existing metrics, relies on comparing generated video samples with corresponding real pairs of the same resolution. Consequently, these metrics are not significantly influenced by variations in resolution. However, metrics like VIS or FVD, which are based on distance or mutual information, do not have an inherent upper bound. While this lack of an upper bound can result in variation in evaluation scale, it is challenging to attribute the observed scale differences as meaningful when comparing evaluation scores among generative models producing images or videos at the same resolution.
>
> **(2) How is the metric affected by the length of videos:**
> STREAM's metric is uniquely designed to be unaffected by the length of videos, unlike VIS and FVD, which rely on video embedders that process a fixed number of frames. STREAM works by aggregating the temporal information from frame-encoded features. It calculates the Fast Fourier Transform (FFT) along the temporal axis using these encoded frame features from an image embedder. This design makes STREAM indifferent to the number of input frames. Furthermore, as the number of processed frames increases, the frequency resolution of STREAM becomes more precise, enhancing the metric's accuracy.
>
> For example, in a 128-frame video with about 5 frames exhibiting spatial or temporal inconsistencies (e.g., visual noise or swapping frames), VIS and FVD (measured for every 16 frames using a sliding window) show significant variations in evaluation values. depending on the sliding window’s stride. Additionally, the average scores calculated in this way is likely to underestimate the actual level of distortions, depending on the stride of the sliding window. In contrast, STREAM, evaluates the entire video signal as a whole, thereby avoiding variations in evaluation values observed in metrics like VIS and FVD. Although we couldn’t report the results in the current version due to insufficient computational resources, we will present these findings in our second version of paper.

---

> > ### Author Response · Authors · 2023-11-22
> >
> > **R3-Q4. Validation of STREAM through experiments on various video resolutions and lengths (Figure 22, 23, 24, and 25):**
> > We are pleased to inform you that we have completed the experiment mentioned by the reviewer. We have conducted experiments to assess STREAM’s performance across different video resolutions and lengths under visual or temporal distortion. We have used gaussian blur (for visual distortion) and local swap (for temporal distortion).
> >
> > **(1) Verification of STREAM on various video resolutions (Figure 22 and 23):**
> > We tested video resolutions of 60$\times$60, 120$\times$120, 180$\times$180, and 240$\times$240. Applying a uniform gaussian blur across resolutions, we noted a more pronounced degradation in lower resolution videos, where fewer pixels carry critical information. This trend is accurately captured by the metrics. In the local swap tests, we shuffled video frames equally across resolutions. Since local swap affects only temporal quality without impacting the spatial quality of the video, an ideal metric should consistently provide low temporal scores as the number of local swaps increases, regardless of resolution. Our results show that FVD and STREAM meet these criteria.
> >
> > **(2) Verification of STREAM on various video lengths (Figure 24 and 25):**
> > We tested video lengths of 16, 32, 64, and 128 frames. When applying a uniform gaussian blur, the spatial quality reduction was consistent across lengths, without temporal quality degradation. Both STREAM and FVD effectively captured this behavior. For local swaps, we used a fixed number of swaps across different lengths. Consequently, local swaps affect longer videos less, as the proportion of unshuffled frames increases. STREAM demonstrated better temporal consistency for longer videos, whereas FVD struggled to reflect this variation. Additionally, FVD exhibited increasing deviations in evaluation scores as the video length increases. This highlights FVD's limitations in evaluating videos longer than 16 frames, as it tends to over or underestimate temporal distortions. Our findings show the credibility of STREAM in handling various video resolutions and lengths, and they provide valuable insights into the limitations of existing metrics in such contexts.

---

### Official Review · Reviewer_1pjF · 2023-11-01

**Soundness:** 3 good
**Presentation:** 2 fair
**Contribution:** 3 good
**Rating:** 6
**Confidence:** 4

**Summary:**

This paper proposed a novel method to evaluate the quality of generated videos. Specifically, it proposed STREAM-T and STREAM-S to assess temporal and spatial aspects of the videos respectively. STREAM-T is designed to measure temporal quality. It evaluates the continuity and consistence of videos by calculating the FFT of real and fake videos and comparing the difference of the frequences. STREAM-S evaluates the spatial quality by classifying the amplitude at frequency of 0 using KNN. Experiments show that the proposed method is able to measure video quality better than current methods such as FVD.

**Strengths:**

1. The paper is well-written and easy to follow.
2. The proposed method proposed a novel method targeting a challenging task, video generation evaluation.
3. Experiments show the effectiveness of proposed method in evaluating temporal quality in video generation.
4. STREAM-T is a reasonable metric since it applies statistical method rather than simple l2 or l1 distance to compute loss of video frequency.

**Weaknesses:**

1. How will the performance be if most of the generated videos are still? Can they be correctly evaluated?
2. Using P&R to compute fidelity and diversity of fake videos is reasonable, but why to use amplitude at frequency of 0 as sample points? I would like to see more explanations from the authors.
3. As video quality is very subjective, a systematic evaluation by human raters is required to compare with current method FVD.
4. The difference in the style of real and fake video datasets may have a huge effect on the result. For example, if real videos tend to have fast changes, the frequency may be concentrated on the high frequency, and the curve fitted may have a higher skewness. In such case, if the curve fitted by fake videos is less steep (fake videos change slowly), skewness may be small, and even if the fake video is of high quality, STREAM-T may be small.

**Questions:**

see weaknessess

---

> ### Author Response · Authors · 2023-11-18
>
> # Reviewer 2
> **R2-W1. Performance of STREAM when most of the generated samples are still:**
> Please see our GC1 in general comments.
> &ensp;
>
> **R2-W2. Reason for using amplitude at frequency 0 when computing STREAM-S:**
> In STREAM-S, we use the amplitude at frequency 0 (i.e., mean amplitude) calculations for two main reasons. First, the mean amplitude encapsulates the average signal information of a video, offering a comprehensive overview of its spatial characteristics. Second,  including amplitudes across all frequencies renders P\&R calculation inaccurate.
>
> More specifically, P\&R is designed to approximate the support of real and fake distributions, typically using the union of spheres centered on data points with radii equal to the distance to the  $k$-nearest neighbor. These spheres are expected to cover at least $k$ data points (or image features) and collectively represent the population data support. However, videos, unlike images, are composed of multiple image frames. For instance, a 16-frame video yields 16 image features obtained by an embedding network. Applying FFT along the temporal axis, we get 8 amplitude features for each video. In this context, using P\&R directly on videos might result in inaccurate data support estimation, as a single sphere may not even encompass all data points from one video. Naturally, in such a scenario, the approximated data support may fail to represent the full scope of video data. To address this, we use the mean amplitude, which represents the average feature of a video, as a single data point. This approach ensures a more accurate and representative approximation of real and fake video data support, enhancing the precision of STREAM-S measurements without limiting it to the number of video frames as shown in our experiments.
> &ensp;
>
> **R2-W3. Compare FVD with STREAM in human evaluation (Appendix A.6):**
> Please see **Section 3.2.2 and Appendix A.6.** We have already compared FVD with STREAM in our human evaluation experiment. For the reviewer’s convenience, summarizing here, we measured Spearman’s correlation between human-evaluation and metrics (STREAM and FVD) in terms of video realism and temporal naturalness. The results show that STREAM has a higher correlation with both aspects evaluated by humans than FVD (i.e.,corr(realism, STREAM) = 0.9, corr(realism, FVD) = 0.7, corr(temporal naturalness, STREAM) = 0.6, corr(temporal naturalness, FVD) = 0.5).
> &ensp;
>
> **R2-W4. Difference in the style of real and fake video datasets may have a huge effect on the result (Table 5):**
> Similar to our response in R2-W2, the objective of video generative models is to mimic the spatial and temporal characteristics of real dataset by estimating the real data distribution. Therefore, when there are “differences in style of real and fake video datasets”, an optimal evaluation metric should be able to discern this difference. Therefore, “when the real videos tend to have fast changes while fake videos change slowly”, STREAM-T should give a small score to reflect this. As suggested by the reviewer, we have compared the real videos with fast FPS and generated videos with slow FPS using STREAM-T **(Table 4 in Appendix A.13)**. In the result, STREAM-T has a decreased value when the generated video has a relatively low FPS compared to the real video, which shows the effectiveness of STREAM-T in capturing the temporal difference.

---

> > ### Comment · Reviewer_1pjF · 2023-11-23
> > **Official Comment by Reviewer 1pjF**
> >
> > I thank authors for providing detailed rebuttal. It addressed most of my concerns. I think evaluating video generative models is a very challenging task. This paper proposed a very interesting idea.
> >
> > Hence, I would like to raise my score to acceptance.

---

### Official Review · Reviewer_QwpP · 2023-11-02

**Soundness:** 3 good
**Presentation:** 2 fair
**Contribution:** 3 good
**Rating:** 6
**Confidence:** 4

**Summary:**

While evaluation metrics for image generative models are relatively comprehensive, those for video are limited. This paper addresses the drawbacks of existing metrics, proposing a new metric, STREAM, which comprehensively evaluates the temporal and spatial aspects of videos. Experimental validation was conducted on several baseline video generative models.

**Strengths:**

- The writing is clear and easy to follow.
- The paper's motivation is strong. Given the rapid development in the field of video generation, traditional metric FVD falls short in representing performance comprehensively. This paper bridges this gap.
- The proposed metric allows independent evaluation of spatial and temporal domains, making it applicable for assessing the quality of long videos.

**Weaknesses:**

- The experimental section of the paper is relatively weak, being limited to a few older baseline models. Authors should focus more on current open-source Text-to-video models, which would provide more convincing results.
- Although UCF-101 is a common benchmark dataset, as a contribution paper introducing a new evaluation metric, testing on a wider range of datasets and tasks, including T2V, video prediction, and the latest works in unconditional generation, would enhance the paper's credibility.

**Questions:**

Despite some experimental limitations, the paper makes substantial theoretical contributions. The problem it addresses holds significant value. Currently, I am providing a borderline accept score, hoping the authors will address the experimental aspects mentioned in my review in their future research.

---

> ### Author Response · Authors · 2023-11-18
>
> # Reviewer 1
>
> **R1-W1. Baseline models are limited to a few older models:**
> We would like to clarify that the unconditional video generative models compared in Section 3.2.2 and 3.2.3 include various recent open-source video models, such as **PVDM (2023), MeBT (2023), TATS (2022), and DIGAN (2022) [1,2,3,4]**. Given the inherent nature of evaluation, achieving a fair comparison between video generation models relies on the use of open-source pretrained models. We kindly request your understanding that many of the recent models showing state-of-art performance often do not make these models publicly available. Please see our answer in R1-W2 for the new experiments we have conducted.
> &ensp; - [1] Yu, Sihyun, et al. "Video probabilistic diffusion models in projected latent space."  Proceedings of the IEEE/CVF Conference on Computer Vision and Pattern Recognition. 2023.
> &ensp; - [2] Yoo, Jaehoon, et al. "Towards End-to-End Generative Modeling of Long Videos with Memory-Efficient Bidirectional Transformers." Proceedings of the IEEE/CVF Conference on Computer Vision and Pattern Recognition. 2023.
> &ensp; - [3] Yu, Sihyun, et al. "Generating videos with dynamics-aware implicit generative adversarial networks." arXiv preprint arXiv:2202.10571 (2022).
> &ensp; - [4] Ge, Songwei, et al. "Long video generation with time-agnostic vqgan and time-sensitive transformer." European Conference on Computer Vision. Cham: Springer Nature Switzerland, 2022.
> &ensp;
>
> **R1-W2. Test a wider range of datasets and tasks:**
> We appreciate your suggestion for a more comprehensive evaluation of our metric. In response,  we conducted two additional experiments: **(1)** Comparison of unconditional generative models (Table 6)  and **(2)** Evaluation of video prediction models (Table 8, Figure 26 and 27). Given the constraints of time and computational resources, we chose to concentrate on these two experiments. Nonetheless, we believe that similar trends would emerge if STREAM were applied to T2V tasks.
>
> **(1) Comparison of unconditional generative models (Table 7):** We have compared unconditional video generative models trained on the Taichi dataset, including state-of-the-art models MeBT [2], TATS [3], DIGAN [4], and LVDM [5]. The results in Table 7 (in Appendix A.16) highlight that, unlike FVD, which aims for model-to-model comparison based on one-dimensional score, STREAM not only reveals the strengths and limitations inherent to each video generative model but also enables multi-dimensional comparison between models.
> &ensp; - [5] He, Yingqing, et al. "Latent video diffusion models for high-fidelity video generation with arbitrary lengths." arXiv preprint arXiv:2211.13221 (2022).
>
> **(2) Evaluation of video prediction models (Table 8, Figure 26 and 27):** (Updated)
> We are pleased to share our added comparison of models across various generative tasks using STREAM. We focused on “video prediction models”, which are currently open-sourced with publicly available pretrained weights: RaMViD [6] and MCVD [7]. For the reviewer’s convenience, we show the results in Table 8 below.
>
> |   |FVD ($\downarrow$)|STREAM-T ($\uparrow$)|STREAM-F($\uparrow$)|STREAM-D ($\uparrow$)|
> |:---|:---:|:---:|:---:|:---:|
> |RaMViD|160.1 ($\pm$2.719)|0.9193 ($\pm$0.0007)|0.5562 ($\pm$0.0151)|0.4709 ($\pm$0.0161)|
> |MCVD|21.63 ($\pm$0.511)|0.9896 ($\pm$0.0002)|0.8355 ($\pm$0.0076)|0.8885 ($\pm$0.0086)|
>
> The FVD score indicates MCVD’s superiority in generating better predictions than RaMViD. STREAM offers a detailed analysis of each model’s performance in terms of fidelity, diversity, and temporal naturalness. While MCVD excels in all aspects, RaMViD lags, particularly in fidelity and diversity.
>
> In our detailed assessment (Appendix A.18, Figures 26 and 27), we observe that  RaMViD tends to produce videos with blurry scenes containing artifacts, which are the source of its low fidelity score. Moreover, it shows temporally inconsistent transitions; e.g., objects in the video tend to suddenly appear or disappear over time. On the other hand, MCVD’s outputs are of higher fidelity and demonstrate smoother, more consistent transitions with fewer discrepancies in object appearance during scene transitions.
>
> Overall, these experiments show the versatility and depth of STREAM in assessing video generative models, enhancing our understanding of their capabilities and limitations.
>
> &ensp; - [6] Höppe, Tobias, et al. "Diffusion models for video prediction and infilling." arXiv preprint arXiv:2206.07696 (2022).
> &ensp; - [7] Voleti, Vikram, Alexia Jolicoeur-Martineau, and Chris Pal. "MCVD-masked conditional video diffusion for prediction, generation, and interpolation." Advances in Neural Information Processing Systems 35 (2022): 23371-23385.

---

### Author Response · Authors · 2023-11-13
**Paper revision plan**

# Paper revision plan
To ensure transparency and demonstrate our ongoing efforts, we’d like to share the revised paper and our plans for future experiments. As we complete each planned item, we will continuously update this content. Detailed responses to reviewer comments will be posted as soon as our experiments and analysis are completed.

Our plan:
- Testing STREAM on the wider range of tasks (R1): **Table 8 (Appendix A.18).**

- Evaluation of latest works in unconditional generations with Taichi dataset (R1): **Table 7 (Appendix A.16).**

- Validate STREAM’s performance when fake videos change slowly (R2): **Table 4 (Appendix A.13).**

- Validate STREAM’s performance with Gaussian blur (R3): **Figure 19 (Appendix A.12).**

- Validate STREAM’s performance with temporally reversed video, spatially flipped video, and both spatially and temporally flipped video (R3): **Table 5 (Appendix A.14).**

- Comparison of metrics when the length and resolution of video change (R3): **Figure 22, 23, 24 and 25 (Appendix A.17).**

- Comparison of metrics with Video Inception Score (VIS) (R4): **Table 1 (Section 3.2.2), Table 2 (Section 3.2.3), Table 5 (Appendix A.14), and Figure 20 and 21 (Appendix A.15).**

- Ablation study on histogram bin size (R4): **Figure 17 and 18 (Appendix A.11).**

- Upload first version of the revised manuscript: **Updated.**

- Upload second version of the revised manuscript: **Updated! (all the changes are highlighted in lime color)**

**We are pleased to inform that our second paper revision has been updated.**

---

### Author Response · Authors · 2023-11-18
**General Comments**

# **General Comments**
We’d like to express our gratitude to all the reviewers for taking time to read and assess our work. Their comments  have been valuable in highlighting  the strengths of our metric—its ability to provide comprehensive assessment of both spatial and temporal aspects. Additionally, their feedback has greatly improved the clarity and understanding of our work.

**Reviewers can find our refinements and additional experiments we conducted in our revised PDF file (all changes are highlighted).**
&ensp;

**GC1. Validation of STREAM when video sample consists of still frames (Figure 4 and 6):**

We have already conducted this experiment in both toy and real data settings in Figure 4 (Section 3.1.3) and Figure 6 (Section 3.2.1). Since **the goal of video generative models is to mimic the characteristics of real dataset by estimating the real data distribution, evaluation metrics must inherently be relative to the characteristics of the reference real dataset.** Therefore, if most of the generated samples are still in a scenario where the real data exhibits temporal changes, a low STREAM-T value is expected. On the contrary, if still videos are dominant in both real and fake datasets, it indicates that the video generative model has accurately estimated the real distribution, leading to a high STREAM-T score. Additionally, in such situations, an effective evaluation would yield a high STREAM-S value if the spatial quality (e.g., fidelity and diversity of video frames) of the generated data is good.
&ensp;

**GC2. Why a multi-dimensional score metric like STREAM is essential (Table 1 and 2):**

**Disentangled metrics, such as our STREAM, are essential for a thorough evaluation of video generation quality.** In response to the reviewers’ comment on the complexity of using multiple sub-metrics in STREAM, we clarify that STREAM is not intended to replace FVD, but rather to complement it by evaluating aspects that FVD might not fully capture. This approach offers a more nuanced guidance for generative models to advance in various quality aspects. As highlighted in Table 1 and 2 of our study, we identified that current video generative models often exhibit a limited diversity in video generation, with a tendency to focus on fidelity. Moreover, we observed a lack of emphasis on temporal aspects of video generative models in Table 2. Therefore, our STREAM highlights the importance of considering not only fidelity but also diversity and temporal aspects when advancing video generative models.

Additionally, disentangled metrics for each quality aspect are essential for a thorough assessment of video generation quality. While human evaluations are reliable for fidelity assessments in 2D images, they fall short in accurately gauging diversity, especially as the complexity of data increases. This limitation becomes more pronounced in videos, where evaluating diversity and temporal naturalness is even more challenging.  FVD, which aggregates all aspects to a single measure, may not effectively capture these nuances. Therefore, STREAM, with its ability to analyze video generative models across various resolutions and aspects, is vital for the field’s progress. It offers a more detailed and aspect-specific evaluation, crucial for the advancement of video generative models.

---

### Meta-Review · Area_Chair_dVXX · 2023-12-05

**Metareview:**

This paper contributes a suite of three metrics for analyzing video generative models. Stream-T for temporal coherence, Stream-F for visual fidelity, and Stream-D for diversity. The former is derived from Fourier features to assess temporal “naturalness”, while the latter two are improvements to Precision & Recall metrics. The motivation for this separation stems from the bias that FVD has towards assessing the spatial quality of the video over other aspects.

Though the initial reviewer scores were recommending rejection, multiple reviewers have increased their score since the rebuttal. Indeed, the reviewers agree that this work addresses an important open problem as FVD is limited of sorts. Further, there is broad agreement that the proposed metrics make sense (and in particular the use of FFT for judging temporal consistency is interesting) and that the empirical evaluation is solid. As part of the rebuttal, the authors were able to provide results for many of the suggested clarification experiments, which further improved the paper.

Nonetheless, even after the reviewer discussion, there is disagreement among the reviewers about the usefulness of the proposed metrics, which is the main contribution. In particular, the Stream suite proposes three new metrics for analyzing video generative models without providing some way to derive a single score. Because of this, it complicates hill-climbing on said metrics for model development. As one reviewer puts it: “if I'm developing a new method and my newest change to the method improves STREAM-F but hurts STREAM-T, is it a good modification or not”. In contrast FVD, albeit it far from perfect as a metric, does provide a single score to optimize on, which is arguably part of the reason for its popularity. The author's response to this critique has been that “ STREAM is not intended to replace FVD, but rather to complement it by evaluating aspects that FVD might not fully capture”. This somewhat sidesteps this issue, though it also weakens the significance of the contribution somewhat.

This paper is borderline, though I don’t think there is much harm in having this paper accepted. In that sense I disagree with reviewer rxEA that this might contribute to greater confusion rather than clarity (though their concerns about usefulness as also pointed out by reviewer srFK stand). If the STREAM metrics are not intended to replace FVD, then they just become additional metrics for understanding (or debugging) how models perform along different dimensions, and not the “main metric” for reporting SOTA performance. I do find that in that case it is important that the authors clarify this further in their paper, i.e. that STREAM is meant to be considered alongside FVD and not as a replacement. Finally, I want to point out that the STREAM metrics have their own advantages over FVD, such as being bounded between 0–1, working regardless of video length, and of course being able to separately assess temporal consistency, visual fidelity and diversity. Moreover, the paper indirectly contributes an interesting study that evaluates FVD along said dimensions.

**Justification For Why Not Higher Score:**

This paper is borderline as it is (see my meta review above), the contribution is not sufficient to warrant a spotlight or oral.

**Justification For Why Not Lower Score:**

Despite concerns about usefulness of the proposed metrics, most of the reviewers agree that there is enough of a contribution to warrant acceptance. In particular, the paper is commended for contributing a very thorough evaluation, addressing and interesting problem, and proposing an interesting set of metrics that themselves offer some advantages vs FVD.

---

### Decision · Program_Chairs · 2024-01-16

Accept (poster)